# Energy stress-induced lncRNA *FILNC1* represses c-Myc-mediated energy metabolism and inhibits renal tumor development

Zhen-Dong Xiao[1], Leng Han [2,3], Hyemin Lee[1], Li Zhuang[1], Yilei Zhang[1], Joelle Baddour[4], Deepak Nagrath[5], Christopher G. Wood[6], Jian Gu[7], Xifeng Wu[7], Han Liang [2,8] & Boyi Gan[1,9,10]

The roles of long non-coding RNAs in cancer metabolism remain largely unexplored. Here we identify *FILNC1* (FoxO-induced long non-coding RNA 1) as an energy stress-induced long non-coding RNA by FoxO transcription factors. *FILNC1* deficiency in renal cancer cells alleviates energy stress-induced apoptosis and markedly promotes renal tumor development. We show that *FILNC1* deficiency leads to enhanced glucose uptake and lactate production through upregulation of c-Myc. Upon energy stress, *FILNC1* interacts with AUF1, a *c-Myc* mRNA-binding protein, and sequesters AUF1 from binding *c-Myc* mRNA, leading to downregulation of c-Myc protein. *FILNC1* is specifically expressed in kidney, and is downregulated in renal cell carcinoma; also, its low expression correlates with poor clinical outcomes in renal cell carcinoma. Together, our study not only identifies *FILNC1* as a negative regulator of renal cancer with potential clinical value, but also reveals a regulatory mechanism by long non-coding RNAs to control energy metabolism and tumor development.

[1] Department of Experimental Radiation Oncology, The University of Texas MD Anderson Cancer Center, 1515 Holcombe Blvd, Houston, TX 77030, USA. [2] Department of Bioinformatics and Computational Biology, The University of Texas MD Anderson Cancer Center, 1515 Holcombe Blvd, Houston, TX 77030, USA. [3] Department of Biochemistry and Molecular Biology, The University of Texas Health Science Center at Houston Medical School, 6431 Fannin St, Houston, TX 77030, USA. [4] Department of Chemical and Biomolecular Engineering, Rice University, 6100 Main Street, Houston, TX 77005, USA. [5] Department of Biomedical Engineering, University of Michigan, Ann Arbor, Michigan 48105, USA. [6] Department of Urology, The University of Texas MD Anderson Cancer Center, 1515 Holcombe Blvd, Houston, TX 77030, USA. [7] Department of Epidemiology, The University of Texas MD Anderson Cancer Center, 1515 Holcombe Blvd, Houston, TX 77030, USA. [8] Department of Systems Biology, The University of Texas MD Anderson Cancer Center, 1515 Holcombe Blvd, Houston, TX 77030, USA. [9] Department of Molecular and Cellular Oncology, The University of Texas MD Anderson Cancer Center, 1515 Holcombe Blvd, Houston, TX 77030, USA. [10] Program of Genes and Development, and Program of Cancer Biology, The University of Texas Graduate School of Biomedical Sciences, Houston, TX 77030, USA. Correspondence and requests for materials should be addressed to B.G. (email: bgan@mdanderson.org)

Cancer cells often exhibit dramatic alterations in energy metabolism and nutrient uptake in order to support their increased proliferation and growth. One major nutrient to support tumor growth is glucose, which can be utilized to generate ATP, the major energy source, as well as to provide carbon source for biosynthetic reactions in cancer cells[1, 2]. Accordingly, extensive studies have shown that energy sensing and metabolism play pivotal roles in cancer biology. For example, AMP-activated protein kinase (AMPK) acts as a critical sensor of cellular energy status. In response to an increase of cellular AMP/ATP ratio caused by glucose deprivation, AMPK is activated and serves to restore energy balance through inhibition of anabolic processes (such as protein or lipid synthesis) and promotion of catabolic processes (such as glycolysis). LKB1, the major upstream kinase required for AMPK activation under energy stress conditions, functions as a tumor suppressor and is frequently mutated in several types of human cancers. Thus, the LKB1–AMPK pathway provides a direct link between energy sensing and tumor suppression[3, 4].

One major catabolic process upregulated in response to energy stress is glycolysis, the metabolic pathway through which the majority of pyruvate metabolized from glucose is converted to lactate. Although normal non-proliferating cells undergo glycolysis only under nonaerobic conditions, most cancer cells mainly rely on glycolysis to generate ATP and building blocks for biosynthetic processes even under aerobic conditions, so called "aerobic glycolysis" or "the Warburg effect"[1]. The glycolysis in cancer cells is regulated by several master transcription factors involved in energy metabolism, most notably the c-Myc transcription factor, the proto-oncogene which is over-expressed in many human cancers. It has been well documented that c-Myc promotes glycolysis through upregulation of various genes involved in glycolysis and energy metabolism[5]. c-Myc expression is tightly controlled under physiological conditions, and the deregulated expression of c-Myc under pathological conditions through various mechanisms (gene amplification, transcriptional activation, and post-transcriptional regulation) results in substantial increase in c-Myc protein levels in cancers, which contributes to tumor development. Indeed, it has been estimated that c-Myc is upregulated in up to 70% of human cancers[6]. Although the regulation of energy sensing and metabolism in cancer development by protein-coding genes has been extensively studied[7], the potential role and mechanism of the more recently identified long non-coding RNAs (lncRNAs) in cancer metabolism remain largely unknown.

Recent advances in the next-generation sequencing technologies have convincingly shown that the human genome encodes a previously unappreciated large number of non-coding transcripts, among which lncRNAs represent a class of transcripts longer than 200 nucleotides and with low protein-coding potential[8, 9]. Although several thousands of lncRNAs have been annotated in the human genome, only a very limited number of lncRNAs have been functionally characterized so far. Current studies on these well-characterized lncRNAs have demonstrated that lncRNAs can function as guides of protein–DNA interactions, scaffolds for protein–protein interactions, decoys to proteins or microRNAs, or enhancers to their neighboring genes[10]. Consistent with these diverse biochemical functions of lncRNAs, lncRNAs have been shown to regulate various biological processes, such as cell proliferation, differentiation, survival, and migration, and its dysregulation impacts on different human diseases, such as cancer and metabolic diseases[11]. However, the specific roles of lncRNAs in energy metabolism and cancer development have remained poorly understood.

Renal cell carcinoma (RCC) makes up ~3% of all adult malignancies and ranks among the top ten cancers in the US[12, 13].

RCC represents a major metabolic cancer type, with significant genetic alterations in several key pathways involved in energy metabolism and nutrient sensing[14]. Using renal cancer as a model system to study cancer metabolism, we previously showed that activation of FoxO transcription factor, a central regulator of tumor suppression and metabolism[15–18], in renal cancer cells led to potent cell cycle arrest and apoptosis induction, which is associated with numerous transcriptional alterations of protein-coding genes[19]. In this study, we further characterize FoxO-regulated lncRNA network in renal cancer, and identify one such lncRNA which, upon energy stress, inhibits c-Myc-mediated energy metabolism and suppresses renal tumor development. Accordingly, this lncRNA is highly expressed in kidney tissue but is downregulated in RCC, suggesting that it functions to repress renal tumor development.

## Results

**Energy stress induces *FILNC1* expression via FoxOs.** Using FoxO(TA)ERT2 stable cell lines, we previously showed that reactivation of FoxO1 or FoxO3 transcription factor by 4OHT treatment (F1 + 4OHT or F3 + 4OHT), compared to vehicle treatment (F1 – 4OHT or F3 – 4OHT), in RCC4 or UMRC2 renal cancer cells induced many transcriptional alterations of protein-coding genes, resulting in cell cycle arrest and apoptosis[19]. Since the arrays used in our expression profiling experiments also contained many probes for lncRNAs, we reanalyzed our transcriptome data sets focusing on lncRNAs. Such analysis indeed identified many potential lncRNAs that are differentially regulated upon the activation of FoxOs, particularly FoxO3 (Fig. 1a). This observation is consistent with our previous finding that the more potent cell growth arrest phenotype was induced by FoxO3 activation in these cells[19]. To study the potential functions of these FoxO-regulated lncRNAs in renal tumor development, we further subjected this list of lncRNAs to computational analysis to identify the lncRNAs which exhibit differential expression in renal tumor compared with normal kidney. Such an effort identified a previously uncharacterized lncRNA that represents one of the most upregulated lncRNAs upon FoxO activation (Fig. 1a) and is downregulated in renal cancer (Fig. 7). We named this lncRNA as *FILNC1* (for FoxO-induced long non-coding RNA 1). The analysis of the genomic locus of *FILNC1* gene revealed that there exist at least three largely overlapping non-coding transcripts from *FILNC1* gene (NR_038399, RP5-899B16.1-001, and RP5-899B16.1-002), which likely represent different splicing isoforms of *FILNC1* gene. Correspondingly, we named these three transcripts as *FILNC1* #1, #2, and #3 (Supplementary Fig. 1). We should mention that, since these transcripts largely overlap, our following experiments on real-time PCR and shRNA-mediated knockdown, or analysis of RNA sequencing (RNA-seq) data from The Cancer Genome Atlas (TCGA) RCC data sets cannot definitively distinguish these three isoforms of *FILNC1* gene, thus the data presented below may represent the concerted functions of these *FILNC1* isoforms.

Quantitative real-time PCR confirmed that *FILNC1* could be potently induced by FoxO3 and moderately upregulated by FoxO1 (Fig. 1b). Since FoxO transcription factors play important roles in metabolic stress response[15, 18, 20], we examined whether *FILNC1* expression can be regulated under any metabolic stress condition. Such analyses revealed that, in different kidney cancer cells, glucose starvation could induce *FILNC1* expression (note that weak induction in RCC4 cells may reflect low-endogenous FoxO expression/activity in this cell line[19]), while other stress conditions, such as glutamine starvation, did not apparently affect *FILNC1* expression (Fig. 1c). Since glucose starvation induces energy stress by depleting ATP, we also tested the effect of other

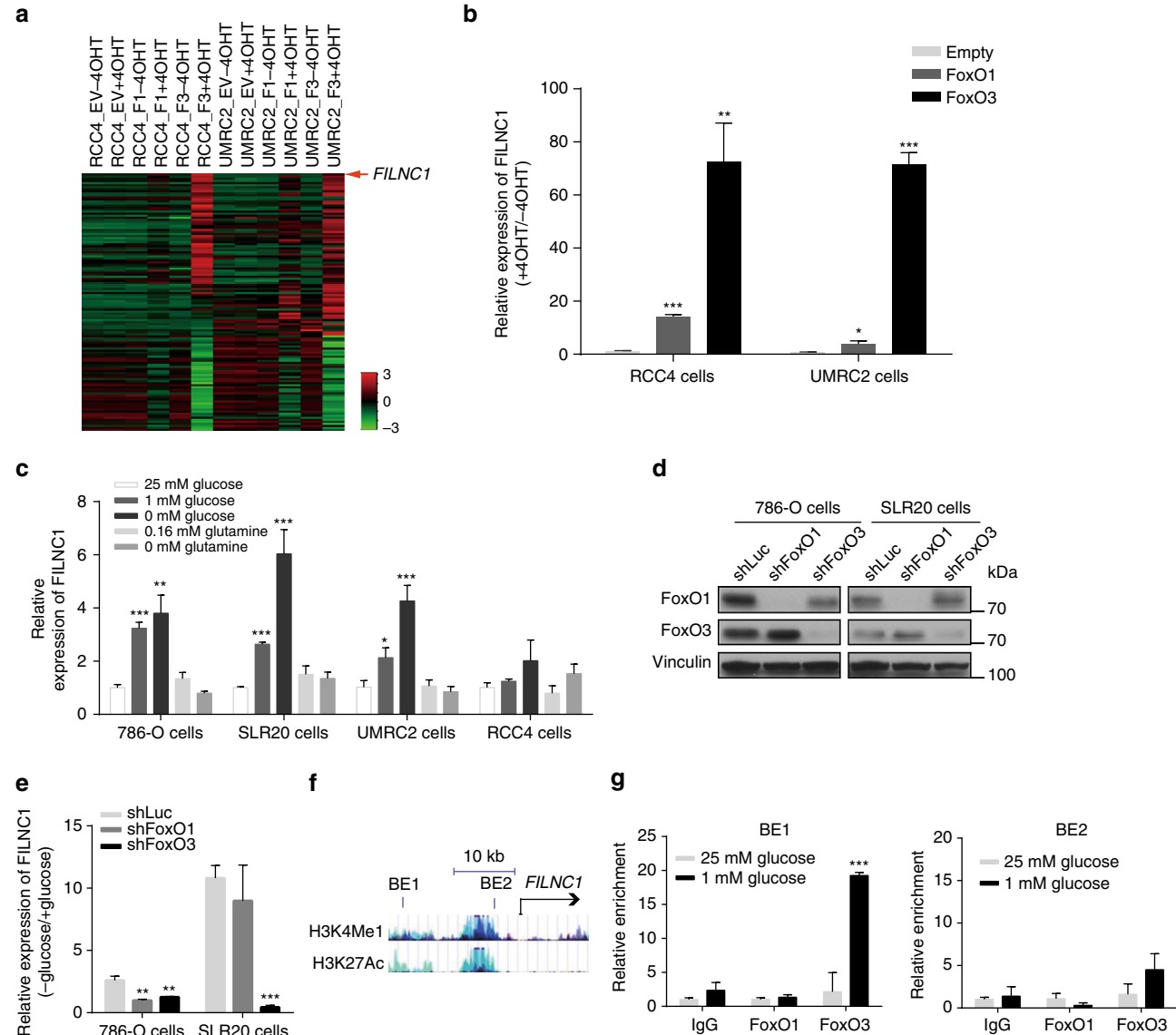

**Fig. 1** *FILNC1* is induced by FoxO activation and glucose starvation. **a** Heatmap of the most differentially regulated lncRNAs from FoxO RCC transcriptome data sets. In each of the two RCC cell lines (RCC4, UMRC2), we generated three stable cell lines: empty vector (*EV*), FoxO1(TA)ERT2 (F1), and FoxO3(TA) ERT2 (F3). 4OHT treatment (+4OHT), but not vehicle treatment (−4OHT), will translocate FoxO1 or FoxO3 from cytoplasm to nucleus and thus activate FoxO-mediated transcription. **b** Bar graph shows the relative expression changes of *FILNC1* by real-time PCR in the indicated cell lines with or without 4OHT treatment for 24 h. **c** Bar graph shows the relative expression changes of *FILNC1* by real-time PCR in renal cancer cells under different culture conditions for 24 h as indicated. **d** Renal cancer cells were infected with different shRNAs, and then subjected to western blotting analysis to examine FoxO1 and FoxO3 expression. **e** Renal cancer cells infected with different shRNAs were cultured in 25 or 1 mM glucose-containing medium for 24 h, and then subjected to real-time PCR analysis to measure *FILNC1* expression. Bar graph shows the relative fold change (−Glucose/+Glucose) of *FILNC1* expression in different cells as indicated. **f** The screenshot shows genome browser tracks for FoxO-binding elements (BE1 and BE2), H3K4Me1 and H3K27Ac profile upstream of *FILNC1* gene. **g** SLR20 cells were cultured in 25 or 1 mM glucose-containing medium for 24 h, and then subjected to ChIP analysis to detect FoxO1/3 binding to FoxO-binding elements identified in *FILNC1* promoter. Bar graph shows the relative enrichment determined by real-time PCR following ChIP analysis using IgG (control), FoxO1, and FoxO3 antibodies. Values represent mean ± s.d. from three independent experiments, two-tailed Student's *t*-test. *$P < 0.05$; **$P < 0.01$; ***$P < 0.001$

energy stress inducers on *FILNC1* expression. Our analysis revealed that treatment of AMP mimetic 5-aminoimidizole-4-carboxamide riboside (AICAR) or the glucose analog 2-deoxy-glucose (2DG) also induced *FILNC1* expression (Supplementary Fig. 2). Importantly, *FoxO* (particularly *FoxO3*) deficiency by shRNA-mediated knockdown attenuated glucose starvation-induced *FILNC1* expression (Fig. 1d, e). Analysis of FoxO-binding element (BE) upstream of *FILNC1* transcription start site revealed that there were two putative FoxO BEs located within the transcription regulatory regions of *FILNC1* gene, which also

correlated with the H3K4me1 and H3K27Ac chromatin modification status profiled by the ENCODE project[21] (Fig. 1f). Chromatin immunoprecipitation (ChIP) assay showed that glucose starvation promoted FoxO3 binding to FoxO BE1, suggesting that *FILNC1* is a direct transcriptional target of FoxO (Fig. 1g). Taken together, our results revealed that *FILNC1* expression can be induced by glucose starvation at least partially through FoxO (mainly FoxO3) transcription factors, well aligned with our previous results on an important role of FoxOs in mediating energy stress response[20].

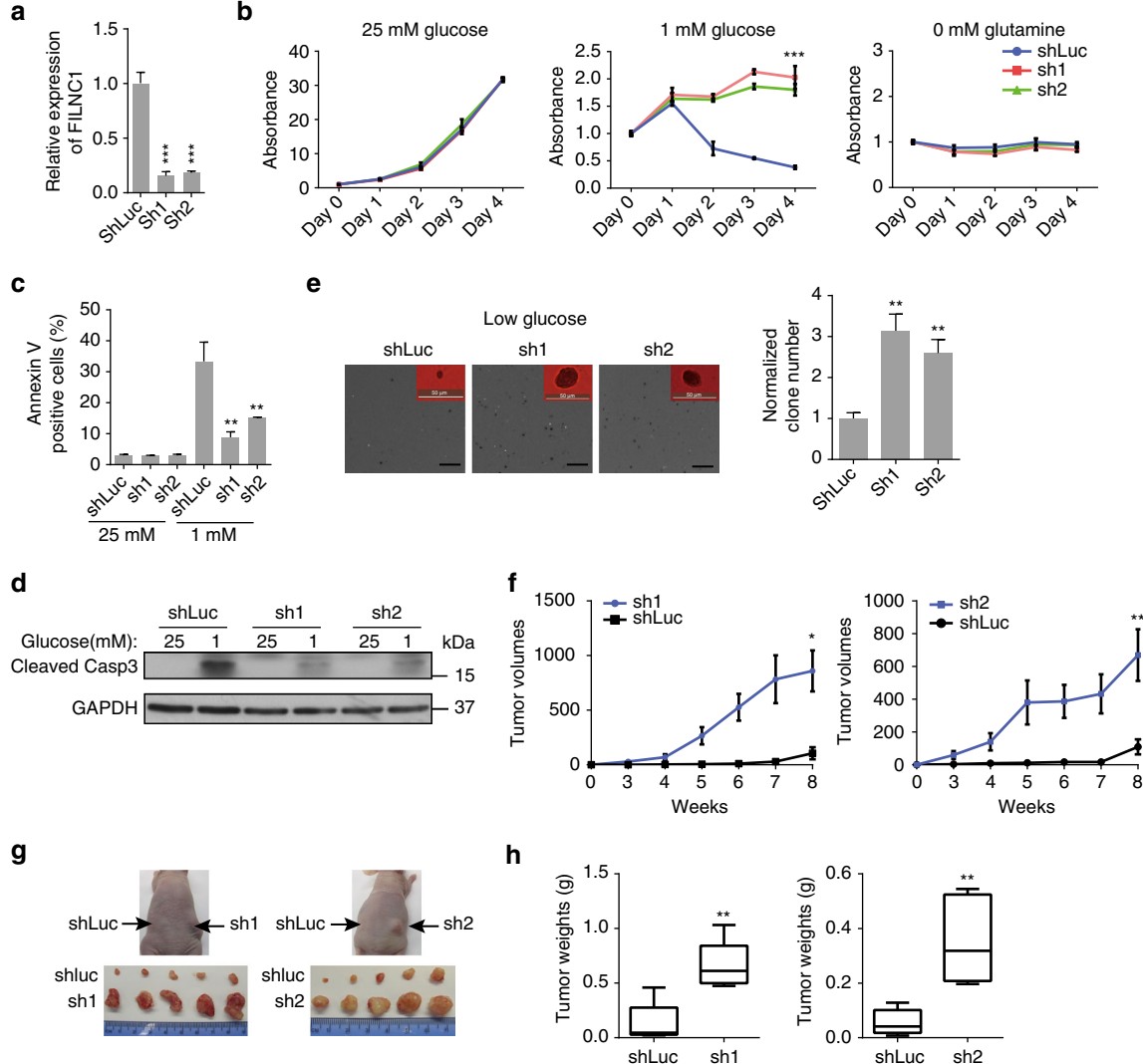

**Fig. 2** *FILNC1* deficiency alleviates energy stress-induced apoptosis and promotes renal tumor development. **a** Bar graph shows *FILNC1* shRNA-mediated knockdown efficiency in 786-O cells. **b** 786-O cells infected with either control shRNA or *FILNC1* shRNA were cultured in various medium for different days as indicated, and then subjected to cell proliferation analysis. **c, d** Control shRNA or *FILNC1* shRNA-infected 786-O cells were cultured in 25 or 1 mM glucose-containing medium for 24 h, then subjected to Annexin V/PI staining followed by FACS analysis to measure the percentages of apoptotic (Annexin V-positive cells/PI-negative cells) or necrotic cells (Annexin V-positive cells/PI-positive cells) (**c**), or to Western blotting analysis to measure Caspase-3 cleavage (**d**). **e** 786-O cells infected with either control shRNA or *FILNC1* shRNA were seeded in soft agar containing 5 mM glucose. The *left images* show the images of representative soft agar colonies (*Scale bars* represent 50 μm and 1.5 mm, respectively). The *right bar graph* shows the relative colony numbers from the soft agar assay (mean ± s.d., *n* = 5 fields per group, each field was assessed from an independent experiment, two-tailed Student's *t*-test). **f** Tumor volumes of 786-O xenograft tumors infected with either control shRNA or *FILNC1* shRNA at different weeks after injection. (mean ± s.e.m, *n* = 5 xenograft tumors, two-tailed Student's *t*-test). **g, h** The representative images (**g**) and tumor weight (**h**), mean ± s.d, *n* = 5 xenograft tumors, two-tailed Student's *t*-test) of different tumor groups at the endpoint. All values, unless otherwise noted, represent mean ± s.d. from three independent experiments, two-tailed Student's *t*-test. *$P < 0.05$; **$P < 0.01$; ***$P < 0.001$

**FILNC1 deficiency inhibits energy stress-induced apoptosis.** The above data prompted further examination of *FILNC1* function in tumor biology and metabolic stress response in renal cancer cells. We identified two independent shRNAs that could potently knockdown *FILNC1* expression in 786-O cells (Fig. 2a). While *FILNC1* knockdown did not affect cell growth under either normal culture, *FILNC1* knockdown increased cell growth under glucose starvation conditions (Fig. 2b). Further analyses revealed that *FILNC1* deficiency did not affect the cell cycle profile (Supplementary Fig. 3), but alleviated cell death induced by glucose starvation as evidenced by both Annexin V staining (Fig. 2c) and cleaved caspase-3 western blotting (Fig. 2d). Correspondingly, *FILNC1* knockdown conferred increased

anchorage-independent growth, as revealed by increases in both colony size and number, under glucose starvation condition (Fig. 2e). In line with the data from in vitro analyses, our in vivo experiments using the xenograft model showed that *FILNC1* deficiency promoted renal tumor development and markedly increased tumor size and weight at the endpoint (Fig. 2f–h). Finally, deficiency of *FILNC1* in UMRC2 cells, another renal cancer cell line, similarly alleviated glucose starvation-induced cell growth suppression in vitro and enhanced renal tumor development in vivo (Supplementary Fig. 4). Collectively, our data strongly suggested that *FILNC1* deficiency protects renal cancer cells from energy stress-induced cell death, resulting in increased anchorage-independent growth and tumor

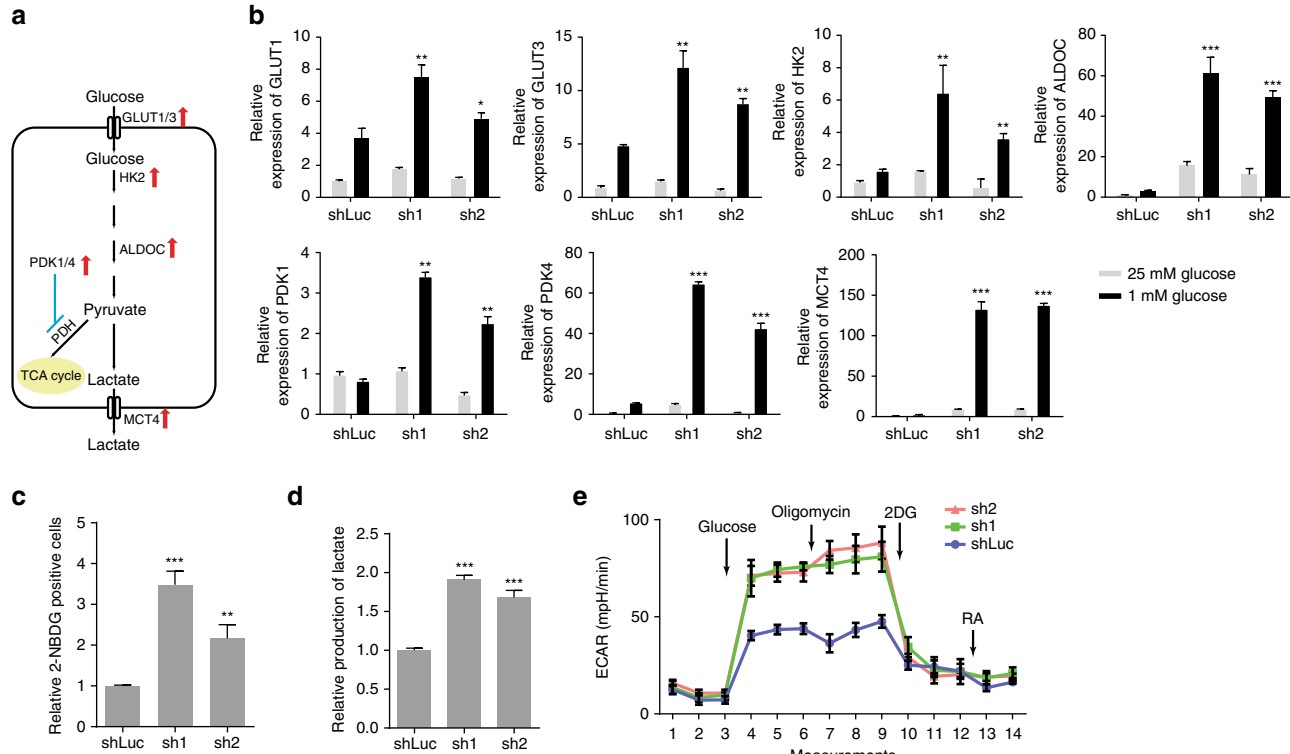

**Fig. 3** *FILNC1* deficiency leads to enhanced glucose uptake and lactate production. **a** The diagram shows the glycolysis pathway and highlights the genes involved in glucose metabolism which are induced by *FILNC1* deficiency under glucose starvation. **b**–**e** 786-O cells infected with either control shRNA or *FILNC1* shRNA were cultured in 1 mM glucose-containing medium for 24 h, and then subjected to various analyses to measure the expression levels of genes involved in glucose metabolism by real-time PCR (**b**), glucose uptake (**c**), lactate production (**d**), or extracellular acidification rate by Seahorse (**e**). Values represent mean ± s.d. from three independent experiments, two-tailed Student's *t*-test. *$P < 0.05$; **$P < 0.01$; ***$P < 0.001$

development in vivo, which is consistent with our data showing the induction of *FILNC1* by FoxO transcription factor, a tumor suppressor in renal cancer[19] (Fig. 1).

**FILNC1 deficiency promotes the Warburg effect**. One key downstream effector in response to energy stress is AMPK[3, 4]. However, *FILNC1* deficiency did not affect the activation status of AMPK or its downstream effectors, such as acetyl CoA carboxylase (ACC) or mammalian target of rapamycin complex 1 (as judged by S6K or S6 phosphorylation levels), under glucose starvation (Supplementary Fig. 5), suggesting it is less likely that *FILNC1* regulates AMPK activation in response to glucose starvation-induced energy stress.

To further study the potential function of *FILNC1* in the regulation of glucose metabolism, we examined whether *FILNC1* deficiency would affect the expression levels of a panel of genes involved in glucose metabolism. Since our functional analyses revealed prominent phenotypes caused by *FILNC1* deficiency mainly under glucose starvation (Fig. 2), here we focused on the effect of *FILNC1* deficiency under glucose starvation condition. Such analyses revealed that *FILNC1* deficiency increased the expression levels of various key regulators involved in glucose uptake, glycolysis, lactate secretion (Fig. 3a, b), under glucose starvation condition. Specifically, under glucose starvation, *FILNC1* knockdown increased the expression levels of several glycolysis genes, including Glut1, Glut3, hexokinase 2, aldolase C (ALDOC), and lactate transporter MCT4, under glucose starvation condition (Fig. 3b and Supplementary Fig. 6). Correspondingly, *FILNC1* deficiency led to increased glucose uptake and lactate production (Fig. 3c, d). Seahorse analysis also revealed increased extracellular acidification rate (ECAR) and no obvious

effect on oxygen consumption rate (OCR) in *FILNC1*-deficient cells (Fig. 3e and Supplementary Fig. 7). *FILNC1* deficiency also increased the expression levels of pyruvate dehydrogenase kinase 1 (PDK1) and PDK4 (Fig. 3b). PDKs phosphorylate and negatively regulate pyruvate dehydrogenase and thus inhibit pyruvate entry into the citric acid cycle. Correspondingly, we showed that *FILNC1* knockdown increased PDK protein levels and PDH phosphorylation (which indicates decreased PDH activity) under glucose starvation (Supplementary Fig. 8A). Pyruvate kinase M isoform switch plays a critical role in cancer metabolism[1, 22]. However, our analysis revealed that *FILNC1* knockdown did not affect PKM1/2 switching (Supplementary Fig. 8B, C). Finally, deficiency of *FILNC1* in UMRC2 cells similarly increased the expression of a panel of glucose metabolism genes, and promoted glucose uptake and lactate production under glucose starvation condition (Supplementary Fig. 9A–C). Taken together, our results revealed that *FILNC1* deficiency results in increased expression of selected genes involved in glucose metabolism under energy stress condition, leading to enhanced glucose uptake and lactate production.

**FILNC1 deficiency increases c-Myc protein level**. Given that *FILNC1* deficiency affects the expression levels of genes involved in glucose metabolism, we reasoned that *FILNC1* may regulate the master transcription factors involved in glucose metabolism, including HIF1α, HIF2α, and c-Myc[5, 7]. Since *HIF1α* is mutated in 786-O cells (the cell line we have used to study *FILNC1* function), it is less likely that *FILNC1* would regulate glucose metabolism through HIF1α. In addition, *FILNC1* knockdown did not affect HIF2α protein level under glucose starvation (Supplementary Fig. 10). On the other hand, *FILNC1* knockdown did not

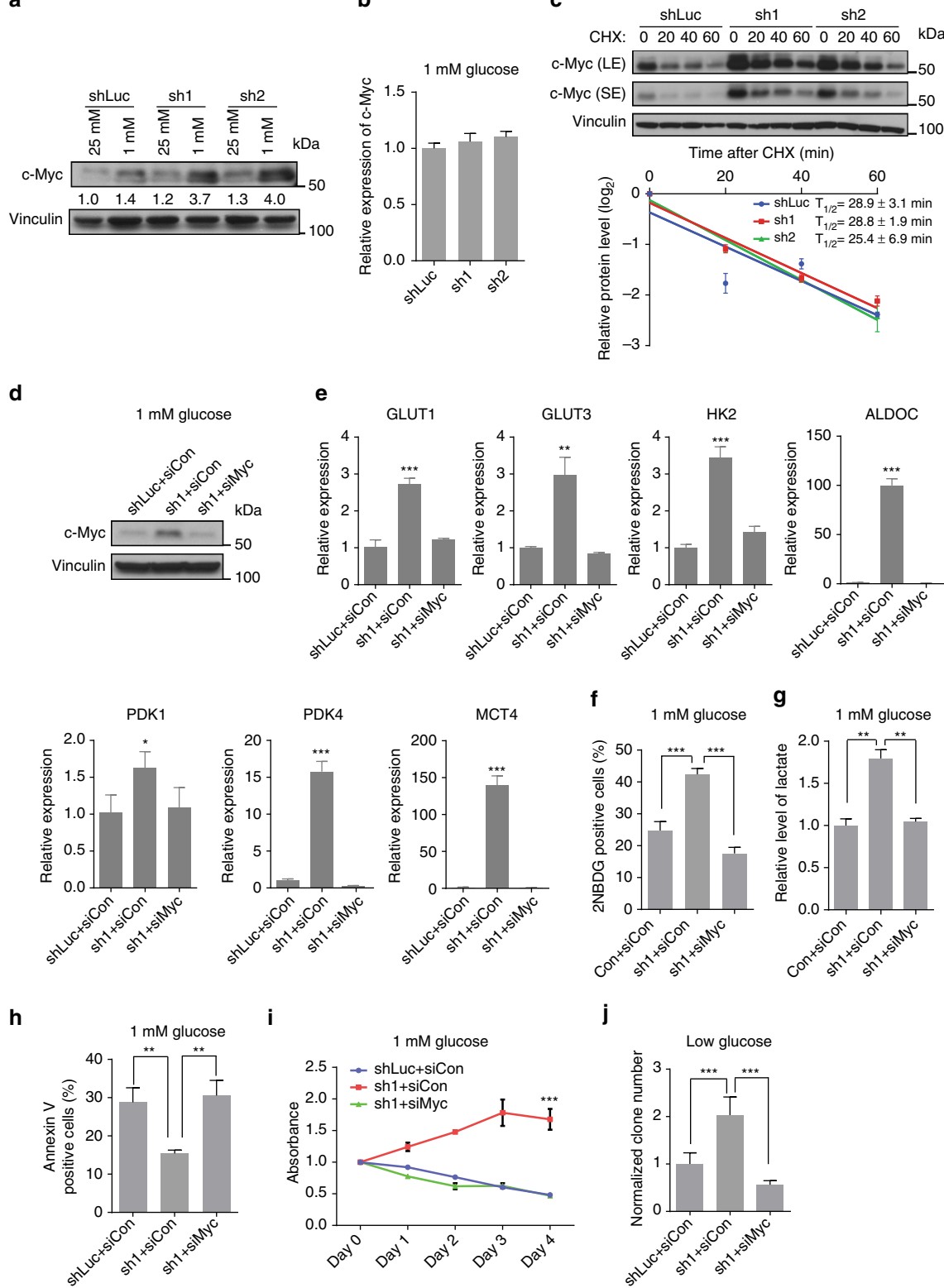

**Fig. 4** *FILNC1* deficiency increases c-Myc protein level. **a** Control shRNA or *FILNC1* shRNA-infected 786-O cells were cultured in 25 mM or 1 mM glucose-containing medium for 24 h, then subjected to western blotting analysis to measure c-Myc protein. **b, c** Control shRNA or *FILNC1* shRNA-infected 786-O cells were cultured in 1 mM glucose-containing medium for 24 h, then subjected to measure c-Myc mRNA using real-time PCR (**b**), or were added with 20 μg/ml Cycloheximide (CHX) for the indicated periods of time, then subjected to western blotting analysis (**c**). **d** Control shRNA or *FILNC1* shRNA-infected 786-O cells were transfected with *c-Myc* siRNA. The cells were cultured in 1 mM glucose-containing medium for 24 h, and c-Myc protein level was detected by western blotting. **e–j** knocking down *c-Myc* in *FILNC1*-deficient 786-O cells represses the expression levels of genes involved in glucose metabolism (**e**), glucose uptake (**f**), lactate production (**g**), cell death (**h**), cell growth (**i**), and colony formation in soft agar (**j**) under glucose starvation condition. Values represent mean ± s.d. from three independent experiments, two-tailed Student's *t*-test. *$P < 0.05$; **$P < 0.01$; ***$P < 0.001$

obviously affect c-Myc protein levels under 25 mM glucose condition, but increased c-Myc protein levels under glucose starvation condition (Fig. 4a for 786-O cells, and Supplementary Fig. 9D for UMRC2 cells). Notably, *FILNC1* knockdown did not affect c-Myc mRNA level or its protein stability under glucose starvation condition (Fig. 4b, c). Finally, *FILNC1* knockdown in

786-O cells did not affect the levels of L-Myc or N-Myc, two other members of Myc family, under glucose starvation condition; indeed, L-Myc or N-Myc exhibited very low expression in the renal cancer cells used in this study (Supplementary Fig. 11A). Consistent with this, a survey of the Cancer Cell Line Encyclopedia (CCLE) expression array data sets revealed that, while c-

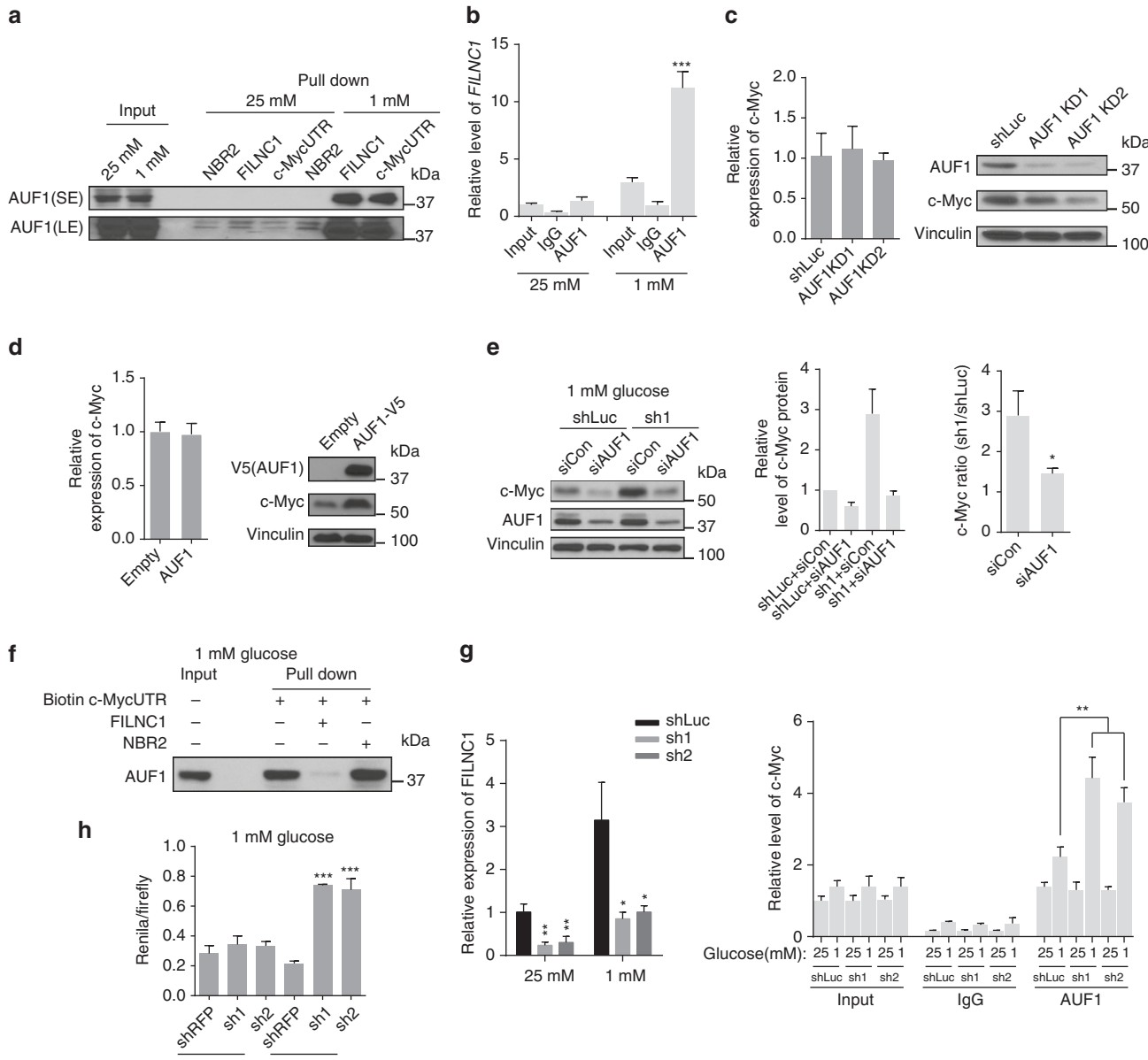

**Fig. 5** *FILNC1* interacts with AUF1 and sequesters AUF1 from binding to *c-Myc* mRNA. **a** In vitro-synthesized biotinylated *NBR2*, *FILNC1*, and *c-Myc* 3′ UTR were incubated with whole-cell lysates from 786-O treated with indicated conditions for 24 h. Precipitation reactions were conducted with streptavidin beads, and then subjected to western blotting analysis for AUF1. **b** Whole-cell lysates from 786-O cells treated with indicated conditions for 24 h were immunoprecipitated with AUF1 antibody or IgG. The levels of *FILNC1* in the precipitates were measured by real-time PCR. **c**, **d** Real-time PCR to measure the mRNA levels of *c-Myc* (*left panel*), and western blotting to measure c-Myc protein level (*right panel*) in 786-O cells infected with *AUF1* shRNAs (**c**) or with *AUF1* cDNA overexpression (**d**). **e** 786-O cells with *AUF1* and *FILNC1* single or double knockdown were cultured in 1 mM glucose-containing medium for 24 h. Whole-cell lysates were collected and subjected for western blotting as indicated. The quantitation of the c-Myc western Blot signal from triplicate experiments and the corresponding ratios were shown as *bar graphs* (*middle* and *right panels* respectively). **f** In vitro-synthesized biotinylated *c-Myc* 3′ UTR were incubated with cell lysates from 786-O cells that had been cultured in 1 mM glucose-containing medium for 24 h. Non-biotinylated *FILNC1* or *NBR2* was used as a competitor in precipitation reactions. After precipitation with streptavidin beads, samples were subjected to western blotting for AUF1. **g** 786-O control or *FILNC1* knockdown (sh1 and sh2) cells treated with indicated conditions for 24 h were subjected to detect *FILNC1* expression using real-time PCR (*left panel*), or were immunoprecipitated with AUF1 antibody or IgG (*right panel*). The levels of *c-Myc* mRNA in the precipitates were measured by real-time PCR. **h** Luciferase activity of a reporter fused to a c-Myc or empty 3′ UTR in 786-O cells treated with indicated conditions for 24 h. Values represent mean ± s.d. from three independent experiments, two-tailed Student's t-test. *P < 0.05; **P < 0.01; ***P < 0.001

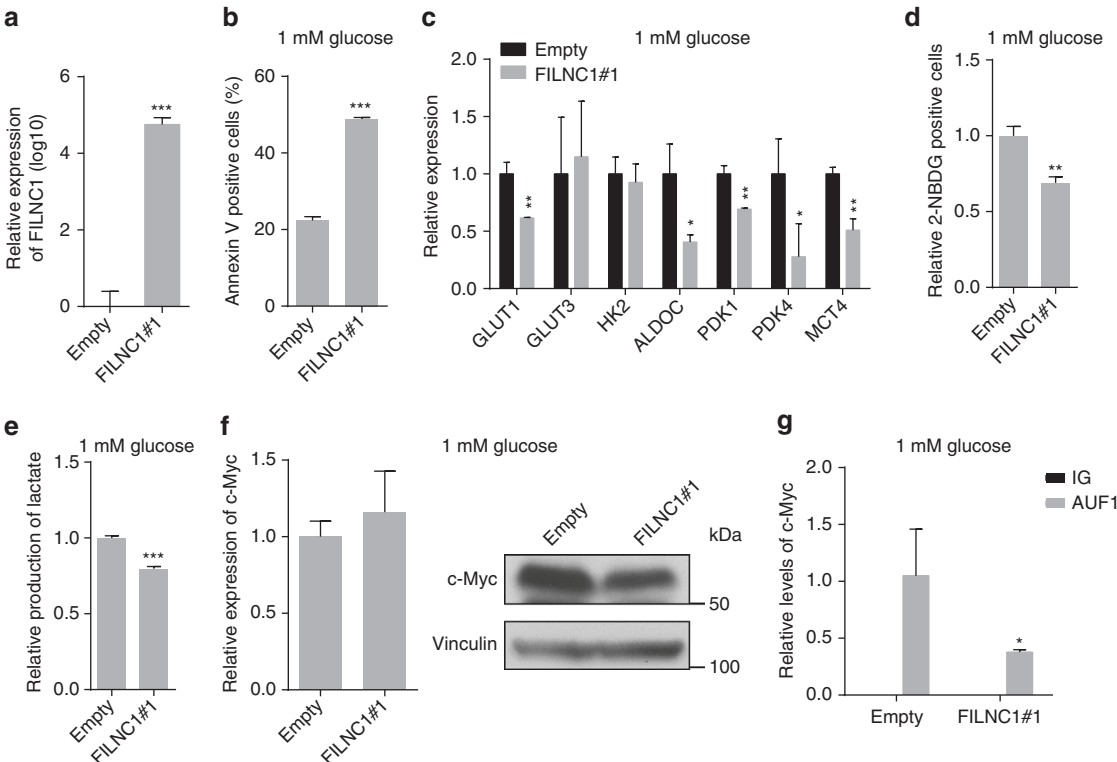

**Fig. 6** The effect of *FILNC1* overexpression on energy metabolism and c-Myc regulation. **a** Bar graph shows the relative expression changes of *FILNC1* by real-time PCR in 769 P cells infected with either empty or *FILNC1* overexpression vectors. **b–e** 769 P cells infected with either empty or *FILNC1* overexpression vectors were cultured in 1 mM glucose-containing medium for 48 h, and then subjected to various analyses to measure the percentages of cell death (Annexin V-positive cells) (**b**), the expression levels of genes involved in glucose metabolism by real-time PCR (**c**), glucose uptake (**d**) and lactate production (**e**). **f** Real-time PCR and western blotting analyses to measure c-Myc mRNA and protein levels in 769 P cells infected with empty or *FILNC1* overexpression vectors which had been cultured in 1 mM glucose-containing medium for 24 h. **g** Whole-cell lysates were collected from 769 P cells infected with either empty or *FILNC1* overexpression vectors that had been cultured in 1 mM glucose-containing medium for 24 h, and immunoprecipitated with AUF1 antibody or IgG. The levels of *c-Myc* mRNA in the precipitates were measured by real-time PCR and normalized using input RNA. Values represent mean ± s.d. from three independent experiments, two-tailed Student's *t*-test. *$P < 0.05$; **$P < 0.01$; ***$P < 0.001$

Myc is highly expressed in many cancer cell lines, L-Myc or N-Myc exhibits low expression in the majority of cancer cell lines, including renal cancer cells[23] (Supplementary Fig. 11B–D). Importantly, analysis of TCGA clear cell RCC (ccRCC, the predominant renal cancer subtype) data sets revealed that c-Myc, but not L-Myc or N-Myc, exhibits higher expression levels in renal cancer than in normal kidney[24] (Supplementary Fig. 11E). Together, these analyses provided strong rationale for our study of c-Myc in the context of *FILNC1* function in renal cancer.

Next, we studied whether c-Myc played any causal role in *FILNC1* regulation of energy metabolism. We knocked down *c-Myc* by siRNA in *FILNC1*-deficient (sh1) cells to the level similar to that in *FILNC1* proficient (shLuc) cells under glucose starvation (Fig. 4d). Notably, *c-Myc* knockdown in *FILNC1*-deficient cells largely normalized the upregulation of glucose metabolism genes, enhanced glucose uptake, and increased lactate production phenotypes caused by *FILNC1* deficiency under glucose starvation condition (Fig. 4e–g and Supplementary Fig. 12), strongly suggesting that *FILNC1* regulates the expression levels of glucose metabolism genes through c-Myc. Further analysis revealed that re-expression of siRNA-resistant *c-Myc* in *FILNC1*/*c-Myc* double knockdown cells normalized the down-regulation of expression levels of glucose metabolism genes caused by *c-Myc* knockdown, confirming the specificity of *c-Myc* siRNA (Supplementary Fig. 13). We should note that the expression changes of some glucose metabolism genes (such as MCT4 and ALDOC) upon *FILNC1* knockdown were much

higher than fold changes of most Myc target genes upon Myc activation. Thus, it is possible that *FILNC1* regulation of the expression of some genes also involves Myc-independent mechanisms. *FILNC1*-deficient cells exhibited decreased apoptosis, increased cell growth and anchorage-independent growth under glucose starvation conditions (Fig. 2). Correspondingly, *c-Myc* knockdown reversed these phenotypes in *FILNC1*-deficient cells (Fig. 4h–j). Together, our data suggested that *FILNC1* suppresses c-Myc levels at the post-transcriptional level, and c-Myc is at least one downstream effector to mediate the biological effects afforded by *FILNC1* deficiency under energy stress.

**FILNC1 sequesters AUF1 from binding to c-Myc mRNA.** To study the potential mechanism(s) by which *FILNC1* regulates the protein levels of c-Myc, we performed RNA-pulldown experiments followed by mass spectrometry to identify *FILNC1*-interacting proteins under glucose starvation condition (see Methods section for detailed description). Such analysis identified a list of potential *FILNC1*-interacting proteins (Supplementary Data 1). In the context of our studies on *FILNC1* regulation of c-Myc, we surveyed this list of potential *FILNC1*-binding proteins and searched for the candidate interacting proteins that have been shown to regulate c-Myc at the post-transcriptional level. Such efforts identified AUF1 as a potential interacting protein of *FILNC1*. AUF1 is an (A + U)-rich elements (AREs)-binding

**Table 1 The expression of *FILNC1* in different types of human cancers.**

| Data set | Compare (vs. normal) | Fold change | P value |
|---|---|---|---|
| Yusenko renal | Chromophobe renal cell carcinoma | **−10.808** | 1.57E−04 |
| | Renal Wilms tumor | **−5.939** | 0.001 |
| | Renal oncocytoma | **−4.256** | 0.026 |
| TCGA breast | Male breast carcinoma | −2.709 | 6.34E−05 |
| | Mixed lobular and ductal breast carcinoma | −1.918 | 0.021 |
| | Mucinous breast carcinoma | −1.825 | 0.005 |
| | Invasive ductal and lobular carcinoma | −1.777 | 0.036 |
| | Invasive lobular breast carcinoma | −1.709 | 2.89E−04 |
| | Invasive ductal breast carcinoma | −1.613 | 1.67E−05 |
| Zhan myeloma | Smoldering myeloma | −2.585 | 0.005 |
| | Monoclonal gammopathy of undetermined significance | −1.6 | 0.021 |
| Sun brain | Anaplastic astrocytoma | −1.869 | 0.01 |
| DErrico gastric | Diffuse gastric adenocarcinoma | −1.842 | 0.009 |
| Tomlins prostate | Prostate carcinoma epithelia | −1.634 | 0.044 |
| Okayama lung | Lung adenocarcinoma | −1.605 | 6.45E−04 |
| TCGA colorectal | Rectal mucinous adenocarcinoma | −1.603 | 2.40E−05 |
| Biewenga cervix | Cervical squamous cell carcinoma | −1.528 | 0.004 |
| Hao esophagus | Esophageal adenocarcinoma | 2.183 | 0.026 |

TCGA, The Cancer Genome Atlas
We examined FILNC1 expression in Oncomine (https://www.oncomine.com) using the following threshold values: *P*: 0.05; fold change: 1.5. Upregulation in tumor samples is designated with a positive fold change, while downregulation in tumor samples is designated with a negative fold change. Bold highlights that *FILNC1* is most downregulated in renal cancers

protein, and it has been shown that AUF1 binds to AREs within 3′ untranslated region (UTR) of *c-Myc* mRNA and promotes *c-Myc* translation without affecting *c-Myc* mRNA level[25]. RNA-pulldown assay using in vitro-synthesized biotinylated RNAs confirmed that glucose starvation increased the interaction of endogenous AUF1 with *FILNC1*, as well as *c-Myc* mRNA, but not with another lncRNA *NBR2*[26–28] (Fig. 5a). Conversely, RNA immunoprecipitation (RIP) assay revealed an enrichment of *FILNC1* in the precipitates of AUF1 compared with IgG control, and glucose starvation further increased the enrichment of *FILNC1* in AUF1 precipitates (Note that glucose starvation resulted in a higher fold increase of the *FILNC1* level in AUF1 precipitates compared with the *FILNC1* input level) (Fig. 5b).

Since mRNA translation mainly occurs in cytoplasm, we examined the subcellular localization of both AUF1 and *FILNC1* in response to glucose starvation. Fractionation experiments revealed that AUF1 and *FILNC1* localized in both cytoplasm and nucleus under normal culturing condition (with 25 mM glucose), and interestingly, glucose starvation increased cytoplasmic localization of both AUF1 and *FILNC1* (Supplementary Fig. 14). As expected, *AUF1* knockdown decreased, while *AUF1* over-expression increased, c-Myc protein levels without affecting the *c-Myc* mRNA level (Fig. 5c, d). Furthermore, knocking down *AUF1* in *FILNC1*-deficient cells compromised the induction of c-Myc proteins caused by *FILNC1* deficiency upon glucose starvation (Fig. 5e), suggesting that *FILNC1* regulates c-Myc at least partially through AUF1. Since AUF1 and *FILNC1* regulate c-Myc oppositely (AUF1 increases, while *FILNC1* suppresses, c-Myc protein level), we reasoned that *FILNC1* may decrease c-Myc protein level through sequestering AUF1 from binding to *c-Myc* 3′-UTR. In support of this hypothesis, in vitro competing RNA-pulldown assay revealed that adding *FILNC1*, but not lncRNA *NBR2*[26, 27], decreased AUF1 binding to *c-Myc* 3′-UTR (Fig. 5f). Correspondingly, RIP assay showed that *FILNC1* deficiency promoted endogenous AUF1 binding to *c-Myc* mRNA under glucose starvation (Fig. 5g). Finally, utilizing luciferase reporter assay in which we fused luciferase reporter genes with *c-Myc* 3′-UTR region, we showed that *FILNC1* knockdown increased luciferase activity under glucose starvation in a *c-Myc* 3′-UTR-dependent manner without affecting the mRNA levels

(Fig. 5h and Supplementary Fig. 15), suggesting that *FILNC1* deficiency upregulates c-Myc protein level at post-transcriptional level under energy stress. Together, these results suggest that *FILNC1* functions as a decoy for AUF1 and decreases c-Myc protein level under glucose starvation condition.

**FILNC1 overexpression inhibits c-Myc and energy metabolism.** To complement with our aforementioned data by loss-of-function approach using shRNA, we also examined the effect of *FILNC1* overexpression in response to energy stress. As discussed previously, there exist at least three largely overlapping splicing isoforms transcribed from *FILNC1* gene locus. Here we chose to focus on *FILNC1* #1 isoform (NR_038399) in our gain-of-function analysis, as *FILNC1* #1 is the longest isoform which covers most of the other two isoforms (Supplementary Fig. 1). Functional analyses revealed that overexpression of *FILNC1* #1 in 769P cells (a renal cancer cell line with low *FILNC1* expression) enhanced glucose starvation-induced cell death (Fig. 6a, b). Under glucose starvation, *FILNC1* #1 overexpression repressed the expression of a subset genes involved in energy metabolism, which was associated with decreased glucose uptake and lactate production in *FILNC1* #1-overexpressing cells (Fig. 6c–e). The expression level of over-expressed *FILNC1* in 769P cells was within reasonable range to that of endogenous *FILNC1* in 786-O cells under glucose starvation (Supplementary Fig. 16), suggesting that the *FILNC1* expression level used in our overexpression studies is of physiological relevance. *FILNC1* #1 overexpression decreased c-Myc protein level, but not mRNA level, and correspondingly, decreased endogenous AUF1 binding to *c-Myc* mRNA under glucose starvation (Fig. 6f, g). Importantly, re-expression of *c-Myc* in *FILNC1* #1-overexpressing cells to the level comparable to that in control cells restored the expression of glucose metabolism genes (Supplementary Fig. 17). Together, our overexpression experiments provide nice complimentary evidence with our knockdown data to further support our conclusion on *FILNC1* function in the regulation of energy metabolism and c-Myc.

**FILNC1 is downregulated in renal cancers.** The aforementioned data prompted us to further examine *FILNC1* expression in

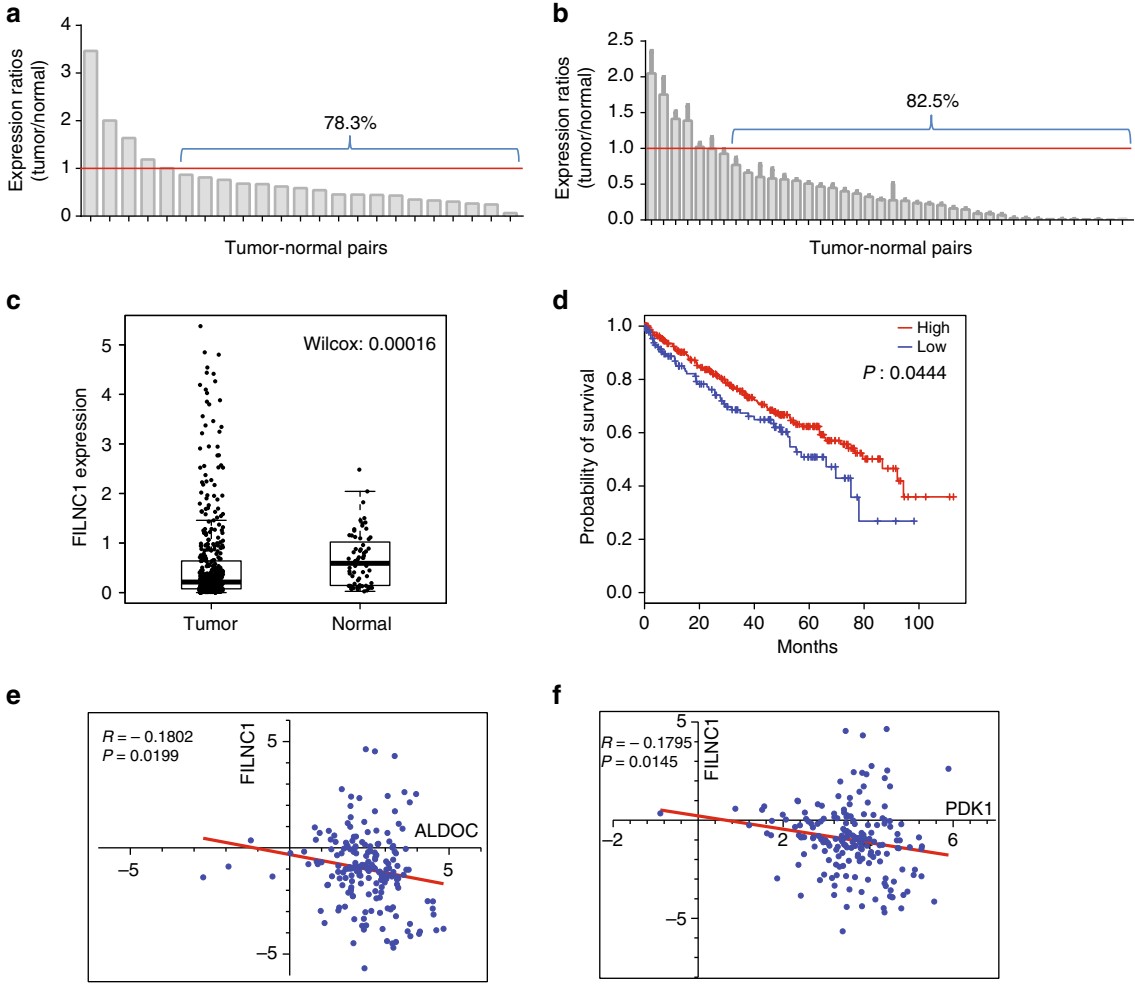

**Fig. 7** *FILNC1* is highly expressed in kidney and its expression is downregulated in renal cancers. **a** Bar graph shows the tumor/normal kidney ratios of *FILNC1* expression in 23 paired ccRCC and normal kidney samples from the data set generated by Peña-Llopis et al.[29] **b** Bar graph shows the tumor/normal kidney ratios of *FILNC1* expression by real-time PCR from 40 matched ccRCC and normal kidney samples. Values represent mean ± s.d. from three independent measures, two-tailed Student's *t*-test. **c** The box plot shows the expression pattern of *FILNC1* for ccRCC and normal kidney samples from the TCGA data set. The *boxes* show the median ± 1 quartile, with whiskers extending to the most extreme data point within 1.5 interquartile range from the box boundaries ($n_{tumor}$ = 449, $n_{normal}$ = 67, Wilcoxon test). **d** A Kaplan–Meier plot of renal cancer patients stratified by the expression levels of *FILNC1* ($n_{high}$ = 312, $n_{low}$ = 134, log-rank test). **e**, **f** Scatter plots show the inverse correlation of *FILNC1* with ALDOC (**e**) or PDK1 (**f**) expression in human renal tumors, respectively

human cancers. We first examined its expression pattern in normal organs/tissues via mining various available public data sets. Such analyses consistently showed that the expression levels of *FILNC1* were higher in kidney than in other organs/tissues (Supplementary Fig. 18). Oncomine expression analysis comparing *FILNC1* expression levels between tumors and corresponding normal tissues revealed that *FILNC1* was most downregulated in renal cancers (Table 1). The analysis of the data set generated by Peña-Llopis et al.[29] revealed downregulation of *FILNC1* expression in the majority of ccRCC samples compared with paired normal kidney samples (Fig. 7a). This conclusion was further confirmed by the examination of *FILNC1* expression in 40 matched normal kidneys and ccRCC samples by real-time PCR (Fig. 7b).

A survey of renal cancer RNA-seq data set from TCGA confirmed downregulation of *FILNC1* expression in renal cancer samples compared with normal kidney samples (Fig. 7c). Kaplan–Meier analysis showed that renal cancer patients with *FILNC1*-low tumors had worse overall survival than those with *FILNC1*-high tumors (Fig. 7d). Consistent with the data that *FILNC1* regulates the expression of glucose metabolism genes

(Fig. 3), computational analyses revealed a negative correlation between *FILNC1* and *ALDOC*, or *PDK1* in renal cancer (Fig. 7e, f). Together, our data showed that *FILNC1* is highly expressed in kidney and downregulated in renal cancers, and that renal cancer patients with low expression of *FILNC1* have poor clinical outcomes, providing further support of our functional data.

## Discussion

Tumor growth requires high energy and nutrient supplies to support its unchecked cell growth. However, such high metabolic potential also brings significant challenge for tumor development: when tumor growth exceeds its energy and nutrient supply, metabolic catastrophe will induce tumor cell apoptosis. Tumor cells often engage strategies of metabolic adaptation to survive the metabolic stress, one of which is to maintain high levels of glycolysis under metabolic stress conditions[30]. c-Myc serves as a master transcription factor to regulate energy metabolism and cell growth, and both its levels and activities must be tightly balanced by various "yin and yang" regulatory mechanisms in normal cells. Accordingly, many cancers cells develop strategies (via various

new genetic alterations) to upregulate c-Myc levels and/or activities to maintain their high rates of glycolysis, resulting in increased tumor development[5]. In this study, we identify an lncRNA-involved regulatory mechanism to control c-Myc levels under energy stress condition. Our findings suggest a model that, in response to energy stress, FoxO transcription factors upregulate the expression of the lncRNA FILNC1. Upon energy stress, FILNC1 interacts with AUF1, and may serve as a decoy to sequester AUF1 from binding c-Myc mRNA, leading to downregulation of c-Myc protein levels. Dysregulation of this regulatory circuitry, such as in renal cancer with decreased FILNC1 expression, leads to increased c-Myc protein levels, enhanced glucose uptake/lactate production and tumor development. Our study thus expands the breath of physiological roles of lncRNA in metabolic stress response and tumor biology.

It has been proposed that aerobic glycolysis in cancer cells serves to redirect glucose flux to other biosynthetic pathways, such as the pentose phosphate pathway or serine/glycine metabolism pathway, for the synthesis of amino acids and nucleotides, and fatty acids, the building blocks for cancer cell growth[1]. Although FILNC1 knockdown did not affect the expression levels of the key genes involved in the pentose phosphate pathway, whether FILNC1 knockdown affects the activity or the flux of the pentose phosphate pathway is unclear. It is also possible that FILNC1 knockdown does not affect the pentose phosphate pathway, but promotes the shunting of glycolytic intermediates into other pathways, such as serine/glycine metabolism pathway. Notably, c-Myc has been shown to regulate serine/glycine metabolism[31]. Our future studies will be directed to examine these interesting questions.

While normal non-proliferating cells oxidize most glucose to $CO_2$ through oxidative phosphorylation, cancer cells tend to convert large fraction of glucose to lactate even under aerobic conditions. Based on the classical view of the Warburg effect, increased rates of glycolysis in cancer cells should be associated with reduced rates of oxidative phosphorylation in the mitochondria (presumably caused by reduced flux from pyruvate into the citric acid cycle in cancer cells). Thus, our data that FILNC1 deficiency increased ECAR but did not affect OCR may seem contradictory to the Warburg effect. However, now we start to appreciate that cancer cells still maintain functional mitochondria, and indeed, mitochondria also play important roles in cancer development[32]. Recent studies revealed that, in certain cancer cells or contexts, reduced flux from pyruvate into the citric acid cycle may also stimulate compensatory oxidation of other metabolites (such as glutamine) to enable persistent citric acid cycle and oxidative phosphorylation function in the mitochondria[33]. Thus, it is possible that FILNC1 deficiency may lead to the reprograming of other metabolic pathways to maintain the citric acid cycle under low glucose condition, which may explain our observation on the effect of FILNC1 knockdown on oxygen consumption. It will be interesting to further examine this hypothesis in the future studies.

Our data show that FILNC1 knockdown only increased c-Myc protein level under glucose starvation condition, and suggest that energy stress regulates FILNC1–AUF1-c-Myc signaling axis via at least two mechanisms. First, energy stress induces FILNC1 transcription, which is at least partly mediated by FoxO transcription factors. Second, energy stress also promotes FILNC1 and AUF1 translocation from nucleus to cytoplasm, and enhances FILNC1–AUF1 interaction. Presumably, these mechanisms amplify the effect of FILNC1 to repress c-Myc levels under energy stress. The exact mechanism by which energy stress regulates FILNC1/AUF1 subcellular localization and FILNC1 interaction with AUF1 remains less clear. It is possible that energy stress induces post-translational modification (such as

phosphorylation) of AUF1, which may affect AUF1 subcellular localization and FILNC1–AUF1 interaction. Alternatively, energy stress may also affect the post-transcriptional modification on FILNC1 (such as RNA methylation)[34], which further influences FILNC1 translocation from nucleus to cytoplasm and its binding to AUF1. It will be interesting to test these hypotheses in the future studies.

Thousands of lncRNAs have been identified and many of them are found dysregulated in human cancers. Some lncRNAs exhibit tissue-specific and/or cancer type-specific expression patterns. In recent years, lncRNAs have been reported to be potential biomarkers in human cancers. For example, HOTAIR was identified as a reliable biomarker for poor prognosis in colorectal cancer[35] and hepatocellular carcinoma[36]. Another notable example is PCA3, a prostate-specific lncRNA which has been used as a biomarker leading to the development of a clinical PCA3 diagnostic assay for prostate cancer diagnosis[37]. As FILNC1 is highly expressed in kidney tissue and is selectively downregulated in kidney cancer, it will be of great interest to examine whether FILNC1 could serve as a biomarker for tissue-of-origin tests and kidney cancer diagnostics.

## Methods

**Cell culture studies**. HEK293T, RCC4, 786-O, and 769P cells were obtained from American Type Culture Collection (ATCC). SLR20 and UMRC2 cells were kind gifts from Dr William Kaelin from Dana Farber Cancer Institute. All cell lines were free of mycoplasma contamination (tested by the vendors and us). No cell lines used in this study are found in the database of commonly misidentified cell lines (ICLAC) based on short tandem repeat profiling performed by vendors. Cancer cell lines with FoxO stable expression were described in our previous publication[19]. Lentiviruses were produced in HEK293T cells with the viral packaging constructs VSVG and Delta 8.9, and used to infect corresponding cells. For glucose or glutamine starvation experiments, cells were cultured in DMEM with different concentrations of glucose (or glutamine) +10% dialyzed FBS[38, 39]. Cell cycle analysis was carried out by PI staining followed by flow cytometry analysis[40, 41]. To measure apoptosis, the cells were stained by Annexin V kit per manufacturer instruction (BD Bioscience) and then subjected to flow cytometry analysis[20, 42]. Cell growth and soft agar assays were conducted as described in our previous publications[19, 43]. Briefly, to examine cell growth, cells were plated in 24-well plates and, at different time points, cells were stained with 0.1% crystal violet (Sigma) for 15 min at room temperature. Stained crystal violet was then extracted with 10% acetic acid. The intensity of the color was measured by a photospectrometry at $OD_{595}$. To examine anchorage-independent growth, 10,000 cells per well in 0.4% agarose on top of a bottom layer of 0.7% agarose were seeded in triplicate wells of 6-well plates. The clones were stained with iodonitrotetrazolium chloride (Sigma) and were counted manually.

**Constructs and reagents**. The primers for constructing shRNAs against FILNC1 and AUF1, and FILNC1 #1 (NR_038399) cDNA are listed in Supplemental Experimental Procedures. shRNAs or cDNAs were subsequently cloned into Lentiviral plasmids pLKO.1-puro or pLVX-puro. c-Myc expression vector was a gift from Dr. Zhimin Lu (the University of Texas MD Anderson Cancer Center). Control siRNA and siRNAs against c-Myc and AUF1 were purchased from OriGene. pENTR-AUF1 cDNA clone was purchased from the Core Facility at MD Anderson Cancer Center and subsequently cloned into pLenti6.2. FoxO1 and FoxO3 shRNAs were described in ref.[19]. 2DG was purchased from Sigma. AICAR was purchased from Toronto Research Chemicals.

**Lactate production and glucose uptake and Seahorse analyses**. To measure lactate production, cells were seeded in 24-well plate in triplicate for 24 h and then refreshed with 1 mM glucose medium overnight. Culture medium was harvested and lactate concentration was detected by lactate test strips and Lactate Plus Meter (Nova Biomedical). Lactate production was normalized by cell protein mass. To measure glucose uptake, cells were seeded in 6-well plate in triplicate for 24 h and then refreshed with 1 mM glucose medium overnight. Culture medium was then removed from each well and replaced with 1 ml of fresh culture medium containing 50 µM fluorescent 2-NBDG (Molecular Probes-Invitrogen). The cells were incubated at 37 °C with 5% CO2 for 30 min. The cells were then washed twice with cold phosphate-buffered saline (PBS) and collected for flow cytometry analysis. ECAR and OCR were measured in the XF96 Analyzer (Seahorse Biosciences) per manufacturer's instructions.

**Quantitative real-time PCR and RIP assay**. Total RNA was extracted from cells using RNeasy (Qiagen) and cDNA was prepared using Superscript II reverse

transcriptase (Invitrogen). Real-time PCR was performed using SYBR Green PCR kit (Invitrogen), and was run on Stratagene MX3000P. RIP was performed with Magna RIP RNA-Binding Protein Immuno-precipitation Kit (Millipore). Briefly, cells were lysed in RIP lysis buffer. Then the lysates were immunoprecipitated with AUF1 antibody or IgG along with protein A/G magnetic beads. RNAs pulled down were purified by phenol chloroform extraction and precipitated in ethanol. cDNA was synthesized and subjected to real-time PCR to detect *FILNC1* or *c-Myc*. The RNA level was normalized with input RNA. The primer sequences used in these assays are described in Supplemental Experimental Procedures.

**RNA pulldown assays and mass spectrometry**. Biotin-labeled RNAs were synthesized by Scientific TranscriptAid T7 High Yield Transcription Kit (Thermo). Cells were lysed, and incubated with biotin-labeled RNAs overnight. The proteins associated with biotin-labeled RNAs were then pulled down with Streptavidin Magnetic Beads (Thermo) after 1 h incubation. The proteins were then washed and used for Western blotting or mass spectrometry (MS) analysis. Cells used in MS analysis were cultured under 1 mM glucose medium for 24 h, and MS analysis included biotinylated *FILNC1* pulldown group, as well as antisense (AS) *FILNC1* and streptavidin beads only pulldown groups as negative controls. In the subsequent computational analysis to enrich true *FILNC1*-binding proteins, we filtered out all the proteins in *FILNC1* pulldown group with less than three spectral count, and set up a cutoff for at least three fold peptide enrichment of *FILNC1* group as compared to either AS *FILNC1* or beads only group. Such analysis generated a list of totally 88 potential binding proteins of *FILNC1*, including AUF1 (Supplementary Data 1). The primer sequences used in these assays are described in Supplemental Experimental Procedures.

**Western blot analysis**. Cultured cells were lysed with NP40 buffer (150 mM sodium chloride, 1.0% NP-40, 50 mM Tris, pH 8.0) containing complete mini protease inhibitors (Roche) and phosphatase inhibitor cocktail (Calbiochem). Western blots were obtained utilizing 20–40 μg of lysate protein. The following antibodies were used in this study: Vinculin (Sigma, V9264, 1:5000 dilution), S6K (Santa Cruz, sc-230, 1:2000 dilution), GAPDH(Cell Signaling Technology, 5174S, 1:2000 dilution), N-Myc(Santa Cruz, sc-791, 1:1000 dilution), L-Myc(Santa Cruz, sc-790, 1:1000 dilution), PDHE1 α (Santa Cruz, sc-377092, 1:1000 dilution), MCT4 (Santa Cruz, sc-50329, 1:1000 dilution), ALDOC(Abcam, ab87122, 1:1000 dilution), pan-PDKs(Abcam, ab115321, 1:1000 dilution), phospho-PDHE1 α (Abcam, ab92696, 1:1000 dilution), AUF1 (Millipore, 03-111, 1:2000 dilution), S6(Cell Signaling Technology, 2217S, 1:1000 dilution), Ser240/244 phospho-S6(Cell Signaling Technology, 5364S, 1:5000 dilution), Thr389 phospho-S6K(Cell Signaling Technology, 9205S, 1:1000 dilution), AMPK α(Cell Signaling Technology, 5832S, 1:1000 dilution), Thr172 phospho-AMPK(Cell Signaling Technology, 2535S, 1:1000 dilution), ACC(Cell Signaling Technology, 3662S, 1:1000 dilution), Ser79 phospho-ACC(Cell Signaling Technology, 3661S, 1:1000 dilution), FoxO1(Cell Signaling Technology, 2880S, 1:1000 dilution), FoxO3(Cell Signaling Technology, 2497S, 1:1000 dilution), PARP(Cell Signaling Technology, 9542S, 1:1000 dilution), cleaved-Caspase 3(Cell Signaling Technology, 9664S, 1:500 dilution), c-Myc(Cell Signaling Technology, 5605S, 1:500 dilution), PKM1(Cell Signaling Technology, 7067S, 1:1000 dilution), PKM2(Cell Signaling Technology, 4053S, 1:1000 dilution). Full size images in main paper are presented in Supplementary Fig. 19.

**Subcellular fractionation**. Cells were collected by trypsin and washed twice with PBS. Cell pellets were lysed in buffer I containing 20 mM HEPES, 10 mM KCL, 2 mM MgCl₂, and 0.5% NP40. After centrifugation, supernatants were collected as cytoplasmic lysis. Pellets were further lysed in buffer II containing 0.5 M NaCl, 20 mM HEPES, 10 mM KCL, 2 mM MgCl₂, and 0.5% NP40. Supernatants were collected as nuclear lysis by centrifugation. Cytoplasmic and nuclear fractions were split for RNA extraction and real-time PCR or protein extraction and Western blotting. Vinculin and PARP were used as markers of cytoplasm and nucleus in Western blotting. GAPDH and U1 were used as markers of cytoplasm and nucleus in real-time PCR. The primer sequences used in these assays are described in Supplemental Experimental Procedures.

**Xenograft model**. All experiments with female athymic Nude Foxn1^nu mice (6-week-old) were performed in accordance with a protocol approved by the Institutional Animal Care and Use Committee of MD Anderson Cancer Center, which is in full compliance with policies of the Institutional Animal Core and Use Committee (IACUC). Mice arriving in our facility were randomly put into cages with at most five mice in each cage. No statistical methods were used to estimate sample size. Approximately $2.5 \times 10^6$ 786-O cells infected with either control shRNA or *FILNC1* shRNA were injected subcutaneously into nude mice. Tumor progression was monitored by bi-dimensional tumor measurements once a week until the endpoint. Mice were sacrificed at the endpoint and the tumors were excised for further experiments. The tumor volume was calculated according to the equation $v = length \times width^2 \times 1/2$. The investigators were blinded to allocation during experiments and outcome assessment.

**Patient samples**. For this study, patients with ccRCC were recruited from the M.D. Anderson Cancer Center. There was no age, gender, ethnicity or cancer stage

restriction on recruitment. All patients provided written informed consent and the study protocol was approved by the M.D. Anderson Cancer Center Institutional Review Board. Tumor and adjacent normal tissues were snap frozen in liquid nitrogen immediately after excision and stored at −80 C. Total RNA was isolated using the mirVana RNA Isolation Kit (Ambion) following standard protocol.

**Computational analysis**. For identification of lncRNAs regulated by FoxO in renal cancer cells, GSE23926 AffymetrixU133-Plus-2.0 data from GEO database[19] were used. All Affymetrix U133-Plus-2.0 probes were matched against Gencode database of lncRNA transcripts. LncRNAs with fold-change > 2.0 or < 0.5 in FoxO activated cells (with 4OHT treatment) were considered as FoxO-regulated lncRNAs. *FILNC1* expression levels in human tissues and in various cancer types were examined by GENEVESTIGATOR and Oncomine data sets respectively.

**TCGA data analysis**. We downloaded the RNA-seq BAM files of TCGA kidney cancer (ccRCC/KIRC) from Cancer Genomics Hub (CGHub, https://cghub.ucsc.edu/), and estimated the expression level of *FILNC1* as Reads Per Kilobase of transcript per Million mapped reads[44]. We used the Wilcoxon rank test to assess the difference between tumor and normal samples. We used the log-rank test to assess the patient survival difference between the high and low expression tumor groups.

**Statistical analysis**. Statistical analyses were performed using GraphPad Prism programme. Sample sizes ($n$) were reported in the corresponding figure legend. No statistical method was used to predetermine sample size. None of the samples/animals was excluded from the experiment. All values were presented as mean ± the standard deviation (s.d.) of at least three independent experiments, unless otherwise noted. Most statistical analyses were performed using two-tailed Student's $t$-test, unless otherwise noted. For all statistical analysis, differences were considered to be statistically significant at values of $P < 0.05$.

**Data availability**. The data supporting the main findings of this study are available within the article and its Supplementary Information and Supplementary Data files. The FoxO-regulated lncRNAs microarray data set referenced in this study are available from the GEO database under accession code GSE23926. The RCC RNA-seq data set of TCGA were obtained from Cancer Genomics Hub (CGHub, https://cghub.ucsc.edu/). Any other data is available from the authors upon request.

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

# ARTICLE

17. Zhang, Y., Gan, B., Liu, D. & Paik, J. H. FoxO family members in cancer. *Cancer Biol. Ther.* **12**, 253–259 (2011).

18. Eijkelenboom, A. & Burgering, B. M. FOXOs: signalling integrators for homeostasis maintenance. *Nat. Rev. Mol. Cell Biol.* **14**, 83–97 (2013).

19. Gan, B. et al. FoxOs enforce a progression checkpoint to constrain mTORC1-activated renal tumorigenesis. *Cancer Cell* **18**, 472–484 (2010).

20. Lin, A. et al. The FoxO-BNIP3 axis exerts a unique regulation of mTORC1 and cell survival under energy stress. *Oncogene* **33**, 3183–3194 (2014).

21. Consortium, E. P. An integrated encyclopedia of DNA elements in the human genome. *Nature* **489**, 57–74 (2012).

22. Christofk, H. R. et al. The M2 splice isoform of pyruvate kinase is important for cancer metabolism and tumour growth. *Nature* **452**, 230–233 (2008).

23. Barretina, J. et al. The Cancer Cell Line Encyclopedia enables predictive modelling of anticancer drug sensitivity. *Nature* **483**, 603–607 (2012).

24. Comprehensive molecular characterization of clear cell renal cell carcinoma. *Nature* **499**, 43–49 (2013).

25. Liao, B., Hu, Y. & Brewer, G. Competitive binding of AUF1 and TIAR to MYC mRNA controls its translation. *Nat. Struct. Mol. Biol.* **14**, 511–518 (2007).

26. Liu, X. et al. LncRNA NBR2 engages a metabolic checkpoint by regulating AMPK under energy stress. *Nat. Cell Biol.* **18**, 431–442 (2016).

27. Liu, X., Xiao, Z. D. & Gan, B. An lncRNA switch for AMPK activation. *Cell Cycle* **15**, 1948–1949 (2016).

28. Liu, X. & Gan, B. lncRNA NBR2 modulates cancer cell sensitivity to phenformin through GLUT1. *Cell Cycle* **15**, 3471–3481 (2016).

29. Pena-Llopis, S. et al. BAP1 loss defines a new class of renal cell carcinoma. *Nat. Genet.* **44**, 751–759 (2012).

30. Jones, R. G. & Thompson, C. B. Tumor suppressors and cell metabolism: a recipe for cancer growth. *Genes Dev.* **23**, 537–548 (2009).

31. Ye, J. et al. Serine catabolism regulates mitochondrial redox control during hypoxia. *Cancer Discov.* **4**, 1406–1417 (2014).

32. Vyas, S., Zaganjor, E. & Haigis, M. C. Mitochondria and cancer. *Cell* **166**, 555–566 (2016).

33. Yang, C. et al. Glutamine oxidation maintains the TCA cycle and cell survival during impaired mitochondrial pyruvate transport. *Mol. Cell* **56**, 414–424 (2014).

34. Li, S. & Mason, C. E. The pivotal regulatory landscape of RNA modifications. *Annu. Rev. Genomics. Hum. Genet.* **15**, 127–150 (2014).

35. Kogo, R. et al. Long noncoding RNA HOTAIR regulates polycomb-dependent chromatin modification and is associated with poor prognosis in colorectal cancers. *Cancer Res.* **71**, 6320–6326 (2011).

36. Yang, Z. et al. Overexpression of long non-coding RNA HOTAIR predicts tumor recurrence in hepatocellular carcinoma patients following liver transplantation. *Ann. Surg. Oncol.* **18**, 1243–1250 (2011).

37. Lee, G. L., Dobi, A. & Srivastava, S. Prostate cancer: diagnostic performance of the PCA3 urine test. *Nat. Rev.* **8**, 123–124 (2011).

38. Dai, F. et al. BAP1 inhibits the ER stress gene regulatory network and modulates metabolic stress response. *Proc. Natl Acad. Sci. USA* **114**, 3192–3197 (2017).

39. Koppula, P., Zhang, Y., Shi, J., Li, W. & Gan, B. The glutamate/cystine antiporter SLC7A11/xCT enhances cancer cell dependency on glucose by exporting glutamate. *J. Biol. Chem.* **292**, 14240–14249, (2017).

40. Gan, B. et al. Lkb1 regulates quiescence and metabolic homeostasis of haematopoietic stem cells. *Nature* **468**, 701–704 (2010).

41. Gan, B. et al. mTORC1-dependent and -independent regulation of stem cell renewal, differentiation, and mobilization. *Proc. Natl Acad. Sci. USA* **105**, 19384–19389 (2008).

42. Lee, H. et al. BAF180 regulates cellular senescence and hematopoietic stem cell homeostasis through p21. *Oncotarget* **7**, 19134–19146 (2016).

43. Lin, A. et al. FoxO transcription factors promote AKT Ser473 phosphorylation and renal tumor growth in response to pharmacological inhibition of the PI3K-AKT pathway. *Cancer Res.* **74**, 1682–1693 (2014).

44. Han, L. et al. The Pan-Cancer analysis of pseudogene expression reveals biologically and clinically relevant tumour subtypes. *Nat. Commun.* **5**, 3963 (2014).

## Acknowledgements

This study has been supported by the Andrew Sabin Family Fellow Award and Institutional Research Grant from MD Anderson Cancer Center, Cancer Prevention & Research Institute of Texas (RP130020), National Institutes of Health (CA181196 and CA190370), Ellison Medical Foundation (AG-NS-0973-13), and Gabrielle's Angel Foundation for Cancer Research (to B.G.). B.G. is an Ellison Medical Foundation New Scholar and an Andrew Sabin Family Fellow. H.L. is supported by the National Institutes of Health (CA143883, CA175486); the R. Lee Clark Fellow Award from The Jeanne F. Shelby Scholarship Fund; a grant from the Cancer Prevention and Research Institute of Texas (RP140462); and the Mary K. Chapman Foundation and the Lorraine Dell Program in Bioinformatics for Personalization of Cancer Medicine. B.G., H.L., and W.X. are members of the M.D. Anderson Cancer Center, and are supported by the National Institutes of Health Core Grant CA016672.

## Author contributions

Z.-D.X. performed most of the experiments with assistance from H.Le, Y.Z., and L.Z.; L.H., and H.Li conducted computational analysis on *FILNC1* expression and status in human cancers. C.G.W., J.G., X.W. provided renal cancer samples. J.B. and D.N. provided help on cancer metabolism-related experiments. B.G. supervised the study. Z.-D.X. and B.G. designed the experiments and wrote the manuscript. All authors commented on the manuscript.

## Additional information

**Competing interests:** The authors declare no competing financial interests.

