## [Peer Review File · Nature Communications]

Reviewers' comments:

Reviewer #1 (Remarks to the Author):

Long non-coding RNAs (lncRNA) are non-protein coding transcripts longer than 200 nucleotides. It is timing and important topic in cancer research. The authors identified a novel lncRNA named FILNC1 (FoxO-induced long noncoding RNA 1), which is regulated by the transcriptional factor FoxO in kidney cancer. FILNC1 inhibition leads to enhanced glucose uptake and lactate production through upregulation of c-Myc protein. The results are interesting and novel to both kidney cancer biology and Myc regulation.

Based on GENCODE annotation, FILNC1 (LOC100132735, NCBI Reference Sequence: NR_038399) is largely overlapped with another lncRNA RP5-899B16.1, which has multiple isoforms:

http://www.ensembl.org/Homo_sapiens/Gene/Summary?db=otherfeatures;g=100132735;r=6:139771073-139860471;t=NR_038399.1

The authors need to define whether these isoforms are also expressed in kidney cancer; if so, whether the siRNAs also target these transcripts.

A full list of the genes identified by RNA-pulldown experiments followed by mass spectrometry should be provided. Is AUF1 the only protein identified?

How about L-Myc and N-Myc? Are they expressed in the cell lines used in this study? Are they regulated by FILNC1?

The overexpression data are important and should be inserted to the main figures but not only presented as an online figure.

Figure 6A is difficult to be read, should be a table.

Reviewer #2 (Remarks to the Author):

The manuscript "FILNC1, an energy stress-induced long non-coding RNA, represses c-Myc-mediated energy metabolism and inhibits renal tumor development" by Xiao et al. delineated how FoxO3 activates FILNC1 which sequesters AUF1 and thereby decreases MYC protein level in ccRCC. It is a very nicely conducted and written study. A few comments are listed for the authors to further strengthen their conclusions.

- 1) Figure 3B. Additional metabolic genes are needed as control. Genes listed at current figure were all up-regulated.
- 2) Figure 3E. Oxygen consumption should be included.
- 3) Figure 4C. MYC half-life should be quantified and plotted.
- 4) Bottom of p.9 c-Myc knock-down significantly "rescued" is confusing, something like "reversed" would be better
- 5) On p.10 "we performed RNA-pulldown experiments followed by mass spec....." Mass spec data need to be included.
- 6) The interaction between FILNC1 and AUF1 is very interesting. Figure S6A-C should include data on 25mM glucose to examine the dynamic relationship.
- 7) Overexpression of AUF1 should be performed for functional assessment, which can help understand how AUF1 and FILNC1 might be regulated.

8) There is a discrepancy between Figures 6D and 6E. In Figure 6D, few tumors are with high FILNC1. In Figure 6E, there are more tumors with high FILNC1. It showed $p = 0.04$. How was the low and high FILNC1 defined?

Reviewer #4 (Remarks to the Author):

FOXO1 and 3 are transcription factors that are commonly deleted in renal cell cancer (RCC). Their re-expression induces many genes involved in cell cycle arrest and apoptosis. In the current work the authors have used two RCC cell lines expressing 4HT inducible FOXO1 or FOXO3 and have identified a lncRNA they term FILNC1 as being highly responsive to FOXO1 and FOXO3 (especially FOXO3). In keeping with others' findings that FOXOs are sensitive to metabolites, they showed FILNC1 transcripts to be negatively regulated by glucose but to have no response to glutamine. Furthermore, glucose starvation stimulated binding of FOXO3 to a FOXO binding site in the FILNC1 promoter. shRNAs against FILNC1 also led to increased cell growth, reduced apoptosis in low glucose medium, increased colony formation and accelerated xenograft tumor growth. Moreover, the authors present evidence that loss of FILNC1 is associated with the induction of the Warburg effect and the up-regulation of Myc that might serve as a master regulator of glycolysis, a supposition that was supported by Myc knockdown studies.

FILNC1 RNA pull-down experiments were performed with the associated proteins being identified by MS. This identified AUF1 as a FILNC1-associated protein and went on to show that AUF knockdown not only decreased FILNC1 transcripts but Myc transcripts as well. The results suggest that FILNC1 functions as a decoy for AUF1 and decreases c-Myc protein level in response to glucose starvation. Finally, the authors are able to show that down-regulation of FILNC1 occurs in a large fraction of primary RCCs with the lowest levels portending a worse prognosis.

Major Comments

Overall, this paper presents a large number of mostly well-done experiments. The manuscript itself is well-written. However, there are a number of questions and concerns that should be addressed in a revised manuscript

1. In discussing Fig. 3, the authors mention that glycolytic genes are up-regulated in response to FILNC1 knockdown but that genes of the pentose phosphate pathway remain unchanged. This seems strange if the purpose of the Warburg effect is to shunt glycolytic intermediates into anabolic pathways, particularly those intended to supply amino acids such as serine and glycine, which use this pathway and nucleotides. They should discuss why the activity of this pathway remains unchanged.

2. The authors use increased levels of PDK1 & 4 transcripts to support the claim that PDH enzymatic activity is decreased. However, to really make this argument, they should show levels of the actual proteins encoded by PDK1 and PDK4. Moreover, PDH (or at least the catalytic E subunit) is also positively regulated by the PDP2 phosphatase whose transcripts AND protein should be decreased upon FILNC1 knockdown. Finally, even if the authors show these results as being consistent with an up-regulation of PDH, they should ultimately show that changes in PDK and PDP2 levels cooperate to increase the overall phosphorylation of PDH and thus reduce its activity. It would be good to demonstrate that this change occurs in vivo as well as in vitro, which could be done with one of their tumor xenografts. Finally, it is not necessary that PDH activity be down-regulated in order to facilitate the Warburg effect. More commonly the glycolytic enzyme proximal to PDH, namely pyruvate kinase (PK) is altered by switching from the PKM1 to the PKM2 isoform, which has a lower K_m for its substrate (PEP) thus allowing for a buildup of upstream glycolytic intermediates. If PDH activity doesn't change, then perhaps PKM1/2 levels are altered.

3. Fig. 3E shows that shRNAs against FILNC1 increase glycolysis in keeping with the Warburg effect. Most Seahorse analyses also include an assessment of oxphos as measured by oxygen

consumption rates. A true Warburg effect should not only be associated with increased rates of glycolysis but should show reduced OCRs as well. This data should be included.

4. AUF was identified by MS as a FILNC1-interacting protein. In these experiments, how many other proteins were identified? The Materials and Methods don't describe the lysate as being pre-cleared with a control biotinylated RNA prior to being used for FILNC1 pulldown. If such a clearing step was not performed, this could have added a significant amount of non-specific background to the experiments.

Minor Comments

1. In Fig. 1A, the text or Fig. legend should mention that BOTH FOXO1 and FOXO3 were induced in each of the two cell lines tested. Otherwise one needs to study the panel very carefully in order to realize this.

2. A section describing MS should be added to the Materials and Methods.

Detailed Point-by-point response to the reviewer's comments:

Reviewer #1 :

Long non-coding RNAs (lncRNA) are non-protein coding transcripts longer than 200 nucleotides. It is timing and important topic in cancer research. The authors identified a novel lncRNA named FILNC1 (FoxO-induced long noncoding RNA 1), which is regulated by the transcriptional factor FoxO in kidney cancer. FILNC1 inhibition leads to enhanced glucose uptake and lactate production through upregulation of c-Myc protein. The results are interesting and novel to both kidney cancer biology and Myc regulation.

We appreciate the positive and insightful comments from this reviewer. We hope that our revision now has fully addressed the critiques from this reviewer.

Based on GENCODE annotation, FILNC1 (LOC100132735, NCBI Reference Sequence: NR_038399) is largely overlapped with another lncRNA RP5-899B16.1, which has multiple isoforms:

http://www.ensembl.org/Homo_sapiens/Gene/Summary?db=otherfeatures;g=100132735;r=6:139771073-139860471;t=NR_038399.1

The authors need to define whether these isoforms are also expressed in kidney cancer; if so, whether the siRNAs also target these transcripts.

We appreciate the reviewer for asking this excellent question. As shown in **rebuttal letter Figure 1**, the analysis of FILNC1 genomic locus revealed that NR_038399 largely overlaps with two other lncRNA transcripts RP5-899B16.1-001 and RP5-899B16.1-002. Since these transcripts are largely overlapped, we propose that they may be different splicing isoforms from the same non-coding gene. Accordingly, the current RNA seq data from TCGA clear cell renal cell carcinoma (ccRCC) datasets cannot clearly distinguish the expression levels of these three transcripts in kidney cancer, and we cannot identify effective shRNA or primers which only targets NR_038399 (that said, the shRNAs or real-time PCR primers used in our study presumably target all three lncRNAs).

Figure 1. The schematic diagram of the genomic region of FILNC1 with different splicing isoforms. Arrows and black boxes represent the direction of transcription and exons respectively.

As more and more RNA seq data are available, we now appreciate that many lncRNAs have multiple isoforms. It is often difficult to study and clearly distinguish the biological functions of all isoforms. In our study, we have also employed gain-of-function approach and showed that overexpression of NR_038399 exerts opposite phenotype to loss-of-function approach by shRNA (see **Figure 6** of our revised manuscript). It is important to note that we choose to study NR_038399 in gain-of-function study because it is the longest isoform among these three largely-overlapped lncRNAs (see **rebuttal letter Figure 1**).

Collectively, given the facts that these three isoforms largely overlap and that RNAseq data or our shRNAs cannot clearly distinguish these three isoforms, in our revised manuscript we would like to revise our definition of FILNC1 gene: we propose that FILNC1 gene contains at least three splicing isoforms (LOC100132735, RP5-899B16.1-001 and RP5-899B16.1-002, now we call them FILNC1 isoform #1, #2, #3, see **rebuttal letter Figure 1**). In our overexpression study, we at least validated the function of FILNC1 #1. We have incorporated this new information into the revised manuscript (see the last several sentences in the first paragraph, page 5 in

our revised manuscript), and the **rebuttal letter Figure 1** now is presented as **Figure S1** in the revised manuscript)

A full list of the genes identified by RNA-pulldown experiments followed by mass spectrometry should be provided. Is AUF1 the only protein identified?

We now provided a full list of the genes identified by RNA-pulldown experiments followed by MS as **Supplementary Table 1** in our revised manuscript (regarding the description of our analysis to identify FILNC1 binding protein, also see our response to the major question 4 from reviewer #4 at page 7 of the rebuttal letter). The MS identified AUF1, as well as other proteins, as potential FILNC1 binding proteins. As described in our revised manuscript (see the last paragraph, page 9), we chose to focus on AUF1, as AUF1 is known to regulate c-Myc translation without affect c-Myc mRNA level, which is consistent with our data on FILNC1 regulation of c-Myc.

*How about L-Myc and N-Myc?
Are they expressed in the cell lines used in this study? Are they regulated by FILNC1?*

We appreciate the reviewer for asking this excellent question. We performed the requested experiment. As shown in **rebuttal letter Figure 2A**, Western blotting analysis showed that L-Myc and N-Myc expression levels are very low in kidney cancer cells used in this study (H526 cell, a lung cancer cell known to express L- and N-Myc, is used as a positive control here), and FILNC1 knockdown did not significantly increase L-Myc or N-Myc expression under low glucose condition (whereas, under the same condition, FILNC1 knockdown significantly increased c-Myc level, see **Figure 4A** in our manuscript).

Consistent with this, a survey of the Cancer Cell Line Encyclopedia (CCLE) expression array datasets revealed that, while c-Myc is highly expressed in many cancer cell lines, L-Myc or N-Myc exhibits low expression in the majority of cancer cell lines, including kidney cancer cells (**rebuttal letter Figure 2B-2D**, red arrows point to kidney cancer cells). Importantly, analysis of TCGA ccRCC datasets revealed that c-Myc, but not L-Myc or N-Myc, exhibits significant higher expression levels in kidney cancer than in normal kidney (**rebuttal letter Figure 2E**). Together, these analyses provide further rationale for our study of c-Myc in the context of FILNC1 function in kidney cancer. We have incorporated these data in our revised manuscript (**Figure S9** and text description in the last paragraph, page 8).

Figure 2. L-Myc and N-Myc in kidney cancer. (A) The expression levels of L- and N-Myc in kidney cancer cells. (B-D) The expression levels of c-Myc, L-Myc, and N-Myc in various cancer types. Red arrows point to kidney cancer. (E) The expression levels of c-Myc, L-Myc, and N-Myc in kidney cancer and normal kidneys.

The overexpression data are important and should be inserted to the main figures but not only presented as an online figure.

We thank the reviewer for this great suggestion. Now we have moved our overexpression data from Figure S8 (in the original manuscript) to **Figure 6** (in our revised manuscript).

Figure 6A is difficult to be read, should be a table.

We now present data in Figure 6A as **Table 1** in our revised manuscript.

Reviewer #2:

The manuscript "FILNC1, an energy stress-induced long non-coding RNA, represses c-Myc-mediated energy metabolism and inhibits renal tumor development" by Xiao et al. delineated how FoxO3 activates FILNC1 which sequesters AUF1 and thereby decreases MYC protein level in ccRCC. It is a very nicely conducted and written study. A few comments are listed for the authors to further strengthen their conclusions.

We appreciate the positive and insightful comments from this reviewer. We hope that our revision now has fully addressed the critiques from this reviewer.

1) Figure 3B. Additional metabolic genes are needed as control. Genes listed at current figure were all up-regulated.

We thank the reviewer for this nice suggestion. Now we have added the expression data from a few other metabolic genes, which showed no expression change upon FILNC1 knockdown under glucose starvation condition (**rebuttal letter Figure 3**, now **Figure 3B** in our revised manuscript).

Figure 3. The relative expression levels of various metabolic genes upon FILNC1 knockdown under glucose starvation condition. *: P<0.05, **: P<0.01, ***: P<0.001.

2) Figure 3E. Oxygen consumption should be included.

We have performed the oxygen consumption experiment as this reviewer suggested. As shown in **rebuttal letter Figure 4** (now **Figure S6** in our revised manuscript), FILNC1 knockdown did not significantly affect the oxygen consumption rate (OCR). Please see our reply to question 3 from reviewer #4 (page 7 of our rebuttal letter) regarding further discussion on our OCR data.

Figure 4. FILNC1 knockdown does not affect oxygen consumption.

3) Figure 4C. MYC half-life should be quantified and plotted.

We thank the reviewer for this nice suggestion. The quantification data is presented in **rebuttal letter Figure 5** (now **Figure 4C** in the revised manuscript), which confirms that FILNC1 knockdown did not significantly affect c-Myc protein half-life. Because FILNC1 knockdown also increased the basal c-Myc protein level, the quantification of the c-Myc levels upon CHX treatment provides further clarification on our conclusion that FILNC1 knockdown did not significantly affect c-Myc protein stability.

4) Bottom of p.9 c-Myc knock-down significantly "rescued" is confusing, something like "reversed" would be better.

We thank the reviewer for the suggestion to further improve the clarity of our manuscript. We now changed the corresponding description from “rescued” to “reversed” as suggested by this reviewer.

5) On p.10 "we performed RNA-pulldown experiments followed by mass spec....." Mass spec data need to be included.

We now provided a full list of the genes identified by RNA-pulldown experiments followed by MS as **Supplementary Table 1** in our revised manuscript (regarding the description of our analysis to identify FILNC1 binding protein, also see our response to the major question 4 from reviewer #4 at page 7 of the rebuttal letter).

6) The interaction between FILNC1 and AUF1 is very interesting. Figure S6A-C should include data on 25mM glucose to examine the dynamic relationship.

Great suggestion from this reviewer! The original Figure S6 shows that, under low glucose condition, AUF1 and FILNC1 mainly localize in cytoplasm. As suggested by this reviewer, we performed the nucleus/cytoplasm fractionation analysis on cells cultured under either 25 or 1 mM glucose condition, and analyzed the localization of AUF1 (by Western blotting) and FILNC1 (by real-time PCR). As shown in **Rebuttal letter Figure 6** (now **Figure S10** in our revised manuscript), the data revealed that glucose starvation increased the cytoplasmic localization of both AUF1 and FILNC1. Note that glucose starvation also increased the expression levels of FILNC1, so the increased cyto/nuc ratio under low glucose condition (from 2.0 under 25 mM glucose condition to 5.3 under 1 mM glucose condition) indicates increased cytoplasmic localization of FILNC1 upon glucose starvation. Together, our data suggest that glucose starvation dynamically regulates AUF1 and FILNC1 subcellular localization, which is also in line with our observation that glucose starvation increased FILNC1-AUF1 interaction (as presumably more FILNC1 and AUF1 are available in the cytoplasm under glucose starvation). In our revised manuscript, we also added a brief discussion on this dynamic regulation (see the second paragraph, page 14).

Figure 5. FILNC1 knockdown does not affect c-Myc protein stability.

Figure 6. Glucose starvation increased cytoplasmic localization of AUF1 (A) and FILNC1 (B). +G: 25 mM glucose; -G: 1 mM glucose.

7) Overexpression of AUF1 should be performed for functional assessment, which can help understand how AUF1 and FILNC1 might be regulated.

In Figure 5C of our manuscript, we showed that AUF1 knockdown decreased c-Myc protein level without affect c-Myc mRNA levels. Now we show that, conversely, AUF1 overexpression increased c-Myc protein level but did not affect c-Myc mRNA levels (**Rebuttal letter Figure 7**, now **Figure 5D** in the revised manuscript). Collectively, both gain-of-function and loss-of-function experiments revealed that AUF1 positively regulates c-Myc levels at post-transcriptional level.

8) There is a discrepancy between Figures 6D and 6E. In Figure 6D, few tumors are with high FILNC1. In Figure 6E, there are more tumors with high FILNC1. It showed $p = 0.04$. How was the low and high FILNC1 defined?

We thank the reviewer for asking this question. The data referred above now correspond to **Figure 7C-7D** in our revised manuscript. We should clarify that both **Figures 7C** and **7D** are from TCGA datasets, but the contents of these two figures are irrelevant. Specifically, in **Figure 7C**, we compared the expression levels of FILNC1 in renal tumors and normal kidney samples, whereas in **Figure 7D**, we focused on renal tumor samples and studied whether FILNC1 expression within renal tumor samples correlates with prognosis of renal cancer patients. We separated tumor samples based on the relative expression level of FILNC1 in tumor samples (for 70% top samples versus 30% bottom samples).

Reviewer #4 : (Remarks to the Author):

FOXO1 and 3 are transcription factors that are commonly deleted in renal cell cancer (RCC). Their re-expression induces many genes involved in cell cycle arrest and apoptosis. In the current work the authors have used two RCC cell lines expressing 4HT inducible FOXO1 or FOXO3 and have identified a lncRNA they term FILNC1 as being highly responsive to FOXO1 and FOXO3 (especially FOXO3). In keeping with others' findings that FOXOs are sensitive to metabolites, they showed FILNC1 transcripts to be negatively regulated by glucose but to have no response to glutamine. Furthermore, glucose starvation stimulated binding of FOXO3 to a FOXO binding site in the FILNC1 promoter. shRNAs against FILNC1 also led to increased cell growth, reduced apoptosis in low glucose medium, increased colony formation and accelerated xenograft tumor growth. Moreover, the authors present evidence that loss of FILNC1 is associated with the induction of the Warburg effect and the up-regulation of Myc that might serve as a master regulator of glycolysis, a supposition that was supported by Myc knockdown studies. FILNC1 RNA pull-down experiments were performed with the associated proteins being identified by MS. This identified AUF1 as a FILNC1-associated protein and went on to show that AUF knockdown not only decreased FILNC1 transcripts but Myc transcripts as well. The results suggest that FILNC1 functions as a decoy for AUF1 and decreases c-Myc protein level in response to glucose starvation. Finally, the authors are able to show that down-regulation of FILNC1 occurs in a large fraction of primary RCCs with the lowest levels portending a worse prognosis.

Major Comments

Overall, this paper presents a large number of mostly well-done experiments. The manuscript itself is well-

Figure 7. AUF1 overexpression increased c-Myc protein levels, but did not affect c-Myc mRNA levels.

written. However, there are a number of questions and concerns that should be addressed in a revised manuscript

We appreciate the positive and insightful comments from this reviewer. We hope that our revision now has fully addressed the critiques from this reviewer.

1. In discussing Fig. 3, the authors mention that glycolytic genes are up-regulated in response to FILNC1 knockdown but that genes of the pentose phosphate pathway remain unchanged. This seems strange if the purpose of the Warburg effect is to shunt glycolytic intermediates into anabolic pathways, particularly those intended to supply amino acids such as serine and glycine, which use this pathway and nucleotides. They should discuss why the activity of this pathway remains unchanged.

We appreciate the reviewer for asking this insightful question. We want to clarify that our data only showed that FILNC1 knockdown did not affect the expression levels (by real time-PCR) of the key genes involved in the pentose phosphate pathway. Whether FILNC1 knockdown affects the activity or flux of the pentose phosphate pathway is unclear and remains to be investigated in the future studies. It is also possible that FILNC1 knockdown does not affect the pentose phosphate pathway, but promotes the shunting of glycolytic intermediates into other pathways such as serine/glycine metabolism pathway. Notably, Myc has also been shown to regulate serine/glycine pathway (for example, see “Serine catabolism regulates mitochondrial redox control during hypoxia. Ye J, et al, Cancer Discovery, 2014”). Our future studies will be directed to further examine these interesting questions. We have now added this discussion in our revised manuscript (see the second paragraph, page 13).

2. The authors use increased levels of PDK1 & 4 transcripts to support the claim that PDH enzymatic activity is decreased. However, to really make this argument, they should show levels of the actual proteins encoded by PDK1 and PDK4. Moreover, PDH (or at least the catalytic E subunit) is also positively regulated by the PDP2 phosphatase whose transcripts AND protein should be decreased upon FILNC1 knockdown. Finally, even if the authors show these results as being consistent with an up-regulation of PDH, they should ultimately show that changes in PDK and PDP2 levels cooperate to increase the overall phosphorylation of PDH and thus reduce its activity. It would be good to demonstrate that this change occurs in vivo as well as in vitro, which could be done with one of their tumor xenografts. Finally, it is not necessary that PDH activity be down-regulated in order to facilitate the Warburg effect. More commonly the glycolytic enzyme proximal to PDH, namely pyruvate kinase (PK) is altered by switching from the PKM1 to the PKM2 isoform, which has a lower Km for its substrate (PEP) thus allowing for a buildup of upstream glycolytic intermediates. If PDH activity doesn't change, then perhaps PKM1/2 levels are altered.

We want to thank this reviewer for providing a very nice background summary on PDH and PDK and its relevance to the Warburg effect. Our analysis in the current study mainly focuses on glucose uptake and lactate production, which can already be explained by differential expression of various genes involved in glucose uptake, glycolysis, and lactate production (including Glut1, Glut3, HK2, ALDOC, MCT4). As suggested by this reviewer, we now measured the levels of PDK and PDH phosphorylation (a biochemical surrogate of PDH activity) by Western blotting (**rebuttal letter Fig. 8A**). Consistent with our real-time PCR data for PDK, FILNC1 knockdown increased PDK protein level and PDH phosphorylation (which indicates decreased PDH activity). We agree with the comment from the reviewer that “it is not necessary that PDH activity be down-regulated in order to facilitate the Warburg effect”, but we hope the reviewer would agree that such data at least provide additional support to our model on FILNC1 regulation of the Warburg effect. In addition, as suggested by this reviewer, we have also examined PKM1/2 switching in our study. Our data revealed that FILNC1 knockdown did not affect PKM1 or 2 levels by either Western blotting or real-time PCR (**Rebuttal**

letter Fig. 8B-8C), suggesting that FILNC1 does not regulate PKM1/2 switch under our experimental conditions. These data now are presented as **Figure S7** in our revised manuscript.

Figure 8. The effect of FILNC1 knockdown on PDK/PDH phosphorylation (A) and PKM1/2 switching (B-C) under 1 mM glucose condition.

3. Fig. 3E shows that shRNAs against FILNC1 increase glycolysis in keeping with the Warburg effect. Most Seahorse analyses also include an assessment of oxphos as measured by oxygen consumption rates. A true Warburg effect should not only be associated with increased rates of glycolysis but should show reduced OCRs as well. This data should be included.

We have conducted the requested experiment. As shown in **rebuttal letter Figure 4** (page 3 of the rebuttal letter, question 2 from reviewer #2), FILNC1 knockdown did not significantly affect oxygen consumption rates. As pointed out by this reviewer, based on the classical view of the Warburg effect, increased rates of glycolysis should be associated with reduced OCR (presumably caused by reduced flux from pyruvate into TCA cycle). On the other hand, now we appreciate that cancer cells often still maintain functional mitochondria, and indeed, mitochondria also play important roles in cancer development (reviewed in “Mitochondria and cancer. Vyas S, et al, Cell, 2016”). Recent studies revealed that, in certain cancer cells or contexts, reduced flux from pyruvate into TCA cycle may also stimulate compensatory oxidation of other metabolites (such as glutamine) to enable persistent TCA cycle and oxidative phosphorylation function in mitochondria (for example, see “Glutamine oxidation maintains the TCA cycle and cell survival during impaired mitochondrial pyruvate transport. Yang C. et al, 2014, Molecular Cell”). Thus, it is possible that FILNC1 deficiency may lead to the reprogramming of other metabolic pathways to maintain TCA cycle and oxidative phosphorylation in mitochondrial under low glucose condition, which may explain our OCR data. It will be interesting to further examine this in the future studies. We now added this discussion in our revised manuscript (the first paragraph, page 14).

4. AUF was identified by MS as a FILNC1-interacting protein. In these experiments, how many other proteins were identified? The Materials and Methods don't describe the lysate as being pre-cleared with a control biotinylated RNA prior to being used for FILNC1 pulldown. If such a clearing step was not performed, this could have added a significant amount of non-specific background to the experiments.

We appreciate this reviewer for asking this important question. As pointed out by this reviewer, a typical MS can detect a large amount of peptides after biotinylated RNA pulldown. In our analysis, we used biotinylated antisense (AS) FILNC1 (AS lncRNA is frequently used as a negative control in lncRNA studies) and streptavidin beads only groups as negative controls. In our subsequent computational analysis, we filtered out all the proteins in FILNC1 pulldown group with less than three spectral counts, and set up a cutoff for at least three fold peptide enrichment of FILNC1 group as compared to either AS FILNC1 or beads only group. Such analysis generated a list of totally 88 potential binding proteins of FILNC1, including AUF1. In the context of our studies of FILNC1 regulation of c-Myc, we then surveyed this list of FILNC1 potential binding proteins, and searched for proteins that have been shown to regulate c-Myc at post-transcriptional level, leading to our focus on AUF1, as AUF1 is known to regulate c-Myc translation without affecting c-Myc mRNA level, which

is consistent with our data with FILNC1 regulation of c-Myc. The list of protein now is provided in **Supplementary Table 1** in our revised manuscript. The detailed description of pulldown assay followed by MS and computational analysis is provided in the Materials and Methods (page 17, under “RNA pull-down assays and mass spectrometry”).

Minor Comments

1. In Fig. 1A, the text or Fig. legend should mention that BOTH FOXO1 and FOXO3 were induced in each of the two cell lines tested. Otherwise one needs to study the panel very carefully in order to realize this.

We thank the reviewer for this nice suggestion. In each of the two RCC cell lines (RCC4, UMRC2), we generated three stable cell lines: empty vector (EV), FoxO1(TA)ERT2 (F1), FoxO3(TA)ERT2 (F3). 4OHT treatment (+4OHT), but not vehicle treatment (-4OHT), will translocate FoxO1/3 from cytoplasm to nucleus and thus induce FoxO-mediated transcription. We now incorporated the this background information into our revised manuscript text (the first paragraph, page 5) and **Figure 1A** legend.

2. A section describing MS should be added to the Materials and Methods.

As suggested by this reviewer, we now added the description of our RNA pull-down assays and mass spectrometry analysis to identify FILNC1 binding proteins in Materials and Methods (page 17, under “RNA pull-down assays and mass spectrometry”).

Reviewers' comments:

Reviewer #1 (Remarks to the Author):

My questions have been addressed. It is ready to be published.

Reviewer #2 (Remarks to the Author):

All comments were adequately addressed to further strengthen the conclusions made in this paper.

Reviewer #4 (Remarks to the Author):

Response to revised manuscript

Overall, the authors have done a good job in responding to most of my concerns as well as those of the other reviewers.

Regarding my suggestion that the authors explore in more detail the regulation and expression of PDH: The fact that PKM1/2 don't change is perhaps even more reason to suspect that the Warburg effect is being mediated at the level of PDH. The ~2-fold increase in pPDH:PDH ratios and the increase in PDK1 in response to FILNC1 knockdown are both consistent with the idea that PDH1 is being inhibited and the post-translational level as I suggested, thereby allowing for the buildup of upstream glycolytic intermediates without the need to evoke changes in PKM1/2. In view of these data that support the authors' hypothesis, it is somewhat disappointing that they did not complete the survey suggested in my comment, namely an evaluation of both PDP2 and an assay for actual PDH activity.

Comments from Reviewer #3 on the previous version of your manuscript.

Xiao et al. present a study of a noncoding RNA identified as being induced by FoxO proteins in renal cancer cells. Although many lncRNAs are induced, they suggest that LOC100132735 (renamed FILNC1) is particularly interesting because it is also induced by the metabolic stress of glucose starvation but not by glutamine starvation. Relatively few lncRNAs have been assigned functions so the initial biological data are potentially interesting. However, I am very skeptical of some of the huge reported changes in RNA levels and even more skeptical of the data surrounding their proposed mechanism of metabolic stress through indirect regulation of MYC protein levels. Specific comments are outlined below.

Fig. 3B: The authors report up to 100-fold changes in mRNA levels in response to the simple depletion of a single lncRNA. These seem huge compared gene regulation changes reported in other studies of lncRNAs. Furthermore, data should be shown for the same shRNAs under +glucose growth and also protein levels for all of the metabolic enzymes characterized.

Fig. 4A. No data on quantitation of protein levels. What is the absolute level of MYC protein +/- shRNA against FILNC under both +glucose and -glucose growth conditions.

Fig. 4C. No quantitation of protein turnover data. Oddly, from the shLuc CHX-chase data, Myc protein levels decrease from 0 to 20 min but then remain virtually constant from 20-60min. In fact, protein loading (vinculin) can account for the apparent decrease at the 60min time point. Even though the protein levels seem higher at the 0min time point in sh1 and sh2, the protein turnover seems significantly faster for sh1 and sh2 compared to shLuc. All of these experiments

need to be carefully quantitated to determine the protein half life under all the conditions.

Fig. 4D: No data shown for RNA and protein levels in cells under normal glucose growth conditions. This difference in protein level seems very small to account for the other measured responses.

Fig. 4F: These data do not seem real based on my extensive experience in RNA quantitation and cell culture in response to MYC. The error bars for virtually all of the measurements seem unnaturally narrow, and the error bars for high measurements for high levels of induction are indistinguishable from low measurements. I am particularly suspicious of the ALDOC and MCT4 data in which genes that are purportedly induced <100-fold with sh1 against FILNC1 and then can be suppressed back to a level virtually indistinguishable from the starting point with only a very modest reduction in MYC (Fig. 4D). As with Fig. 3B, protein levels should be determined in addition to RNA.

Fig. 5A: AUF1 has been described in many studies as a MYC mRNA binding factor, including the Liao/2007 paper that the authors reference. In particular, Liao/2007 identified MYC mRNA bound to AUF1 in a RIP experiment, and previous studies showed an interaction between AUF1 and MYC mRNA using in vitro binding experiments. Furthermore, the methods for the current study state that cell lysates were incubated with biotinylated RNA overnight, then RNA-protein complexes were isolated and subjected to mass spec. Were these experiments conducted under specific conditions of glucose starvation? Given the published data on AUF1 interaction with the MYC UTR and the protocol described in the manuscript, there is no explanation for why the authors do not see an interaction between the MYC UTR and AUF1 in vitro in the +glucose lane (Fig. 5A, lane 5). This is very contrary to published data. Furthermore, the supposed differential binding between +glucose and -glucose extracts is too striking to be believable (Fig. 5A: compare lanes 4 and 5 to lanes 7 and 8). Finally, why does AUF1 bind to FILNC1 with +glucose in Fig. 5B but not 5A? These data don't make sense.

Fig. 5B: The authors state that the data is presented for AUF binding to FILNC1 after normalization of FILNC1 input level. I don't understand the logic in normalizing. If there's more FILNC1 bound to AUF per cell, then these data should be presented along with quantitation of RNA levels.

Fig. 5D: To this reviewer's eye, the ratio of lanes 1 and 2 (siCon/siAUF1) is the same as lanes 3 and 4 (siCon/siAUF1 with additional sh1 against shRNA FILNC1). Why are no data shown for +glucose? If AUF has a constant impact on MYC protein level regardless of cell state, there may be no specificity to FILNC. See comments on Fig. 5F below.

Fig. 5F: The experiment in this figure offered the authors an opportunity to directly test their proposed model, i.e. that FILNC1 titrates AUF protein away from the MYC mRNA and hence suppresses MYC expression. This figure should have presented a straightforward assessment of the amount of MYC mRNA bound to AUF1 in a RIP experiment +/- glucose, with careful quantitation of all mRNA and protein levels, including FILNC1. These data should also be presented with careful normalization to levels per cell. Instead, the authors present a figure with a complex set of binding ratios that obscure the actual level of RNA-protein binding. This presentation is both confusing and illogical, and it does not directly address their model.

Fig. 6 presents complex correlative data and the results are barely within the range of statistical significance.

Detailed Point-by-point response to the reviewer #3's comments based on the previous version of manuscript:

Our original manuscript was reviewed by reviewers #1, 2, and 4 with favorable comments. After the previous revision, all three reviewers have supported the publication of our manuscript in *Nature Communications*. Somehow a late report from the reviewer #3 based on our original version of manuscript was not sent to us initially, but was sent to us after the first round of revision (with the support from all three other reviewers). Here we tried our best to address the remaining critiques from the reviewer #3, some of which had already been addressed in the previous revision. Since the reviewer #3 did not participate in the previous revision, here we also attached the previous rebuttal letter after this rebuttal letter, so that the reviewer #3 could have a better view of our revision.

Xiao et al. present a study of a noncoding RNA identified as being induced by FoxO proteins in renal cancer cells. Although many lncRNAs are induced, they suggest that LOC100132735 (renamed FILNC1) is particularly interesting because it is also induced by the metabolic stress of glucose starvation but not by glutamine starvation. Relatively few lncRNAs have been assigned functions so the initial biological data are potentially interesting. However, I am very skeptical of some of the huge reported changes in RNA levels and even more skeptical of the data surrounding their proposed mechanism of metabolic stress through indirect regulation of MYC protein levels. Specific comments are outlined below.

Fig. 3B: The authors report up to 100-fold changes in mRNA levels in response to the simple depletion of a single lncRNA. These seem huge compared gene regulation changes reported in other studies of lncRNAs. Furthermore, data should be shown for the same shRNAs under +glucose growth and also protein levels for all of the metabolic enzymes characterized.

We thank the reviewer for asking this question. As suggested by this reviewer, here we show mRNA level changes of the metabolic enzymes in control shRNA vs FILNC1 shRNA-infected cells under both 25 mM and 1 mM glucose conditions (rebuttal letter **Figure 1**, now Figure 3B in our revised manuscript). The data reveal that FILNC1 knockdown significantly increased the expression levels of various metabolic enzymes under glucose starvation condition, with no or moderate effect under normal culture conditions, which is also consistent with the effect of FILNC1 knockdown on Myc level (rebuttal letter **Figure 2**, see below). MCT4 and ALDOC, the two genes with 50-100 fold changes upon FILNC1 knockdown, likely exhibit very low expression levels in control shRNA cells, as judged by high Ct values from real time PCR analysis; under the backdrop of this low basal expression, we consistently observed that FILNC1 knockdown significantly increased the expression levels of these two genes (by decreasing 6-7 Ct values in real time PCR analysis). Thus, the dramatic fold changes of these two genes upon FILNC1 knockdown likely reflect the low basal expression of these genes in control shRNA cells.

Figure 1. The effect of FILNC1 knockdown on the expression levels of various metabolic enzymes under 25 and 1 mM glucose. *: P<0.05; **: P < 0.01; *: P<0.001.**

In our previous revision, we already show that FILNC1 knockdown also increased PDK protein level under glucose starvation (see rebuttal letter **Figure 2**, Figure S7A in the revised manuscript. Note here we used pan-PDK antibody that recognizes different PDK isoforms, so the increase shown here is likely to be under-estimated). We hope the reviewer would agree that our current data are sufficient to draw our major conclusion, and Western blotting of all metabolic enzyme proteins be unnecessary.

Figure 2. The effect of FILNC1 knockdown on PDK and PDH phosphorylation under 1 mM glucose condition.

Fig. 4A. No data on quantitation of protein levels. What is the absolute level of MYC protein +/- shRNA against FILNC under both +glucose and -glucose growth conditions.

We thank the reviewer for asking this question. Here we presented the data showing the absolute level of Myc protein in control shRNA vs FILNC1 shRNA-infected cells under both 25 mM and 1 mM glucose conditions with quantification of protein levels (rebuttal letter **Figure 3**, now Figure 4A in the revised manuscript). The results show that FILNC1 knockdown did not obviously affect Myc level under normal culture condition (with 25 mM glucose), but significantly increased Myc level under glucose starvation condition. Since FILNC1 did not obviously affect Myc level under normal culture condition (nor affected cell growth under normal culture condition, see Figure 2 of our manuscript), in subsequent analyses, such as further analysis on whether FILNC1 knockdown affects Myc protein degradation (rebuttal letter **Figure 4**), or the effect of Myc knockdown in FILNC1 deficient cells (rebuttal letter **Figure 6**), we focus on glucose starvation condition.

Figure 3. FILNC1 knockdown increases Myc protein level under glucose starvation.

Here we also want to address two potential questions that may arise from the reviewer based on this data.

First, based on the data that glucose starvation induced FILNC1 expression and that FILNC1 knockdown increased Myc level, one simple prediction would be that glucose starvation should decrease Myc level. However, under our experimental condition we actually observed that glucose starvation did not significantly change, or even moderately increased, Myc level in control shRNA-infected cells (and FILNC1 knockdown further increased Myc level under glucose starvation condition). We propose that Myc level must be tightly balanced by various positive and negative regulatory mechanisms under glucose starvation condition. Thus, glucose starvation-induced Myc level may be controlled by other mechanisms. In any case, our data clearly show that FILNC1 is at least one negative regulator of Myc under glucose starvation, as FILNC1 deficiency increased Myc level under glucose starvation.

Second, why does FILNC1 knockdown mainly affect Myc level under glucose starvation condition? We suggest at least two mechanisms account for such glucose starvation-induced regulation: glucose starvation induces FILNC1 expression through FoxO transcription factors, and glucose starvation also increases FILNC1 binding to AUF1 and thus sequesters AUF1 from binding Myc mRNA.

Fig. 4C. No quantitation of protein turnover data. Oddly, from the shLuc CHX-chase data, Myc protein levels decrease from 0 to 20 min but then remain virtually constant from 20-60min. In fact, protein loading (vinculin) can account for the apparent decrease at the 60min time point. Even though the protein levels seem higher at the 0min time point in sh1 and sh2, the protein turnover seems significantly faster for sh1 and sh2 compared to shLuc. All of these experiments need to be carefully quantitated to determine the protein half life under all the

conditions.

In our previous revision, reviewer #2 also suggested us to quantify Myc protein half life shown in Figure 4C (see question 3 from reviewer #2). Here we show the same quantified data that has been incorporated into our previous revised manuscript (shown here as rebuttal letter **Figure 4**), which confirms that FILNC1 knockdown did not significantly affect c-Myc protein half-life under 1 mM glucose condition. Because FILNC1 knockdown did not significantly affect Myc protein level under normal culture condition (with 25 mM glucose), we did not go on to examine whether FILNC1 knockdown would affect Myc protein half life under normal glucose condition.

Fig. 4D: No data shown for RNA and protein levels in cells under normal glucose growth conditions. This difference in protein level seems very small to account for the other measured responses.

Figure 4. FILNC1 knockdown does not affect c-Myc protein stability under 1 mM glucose condition.

Figure 5. Knockdown Myc in FILNC1 Knockdown cells under 1 mM glucose condition.

This question refers to the data shown in rebuttal letter **Figure 5**. As we explained in response to Figure 4A, since FILNC1 knockdown mainly affects Myc level (as well as other biological effects) under glucose starvation condition, here we focus on Myc level under glucose starvation condition. The data in this figure showed that (1) FILNC1 (sh1) knockdown increased Myc level under glucose starvation, and (2) we then knocked down Myc by

siRNA in FILNC1 deficient (sh1) cells to the level similar to that in FILNC1 proficient (shLuc) cells under glucose starvation. We hope the reviewer would agree that the data carry sufficient information for our proposed experiments.

Fig. 4F: These data do not seem real based on my extensive experience in RNA quantitation and cell culture in response to MYC. The error bars for virtually all of the measurements seem unnaturally narrow, and the error bars for high measurements for high levels of induction are indistinguishable from low measurements. I am particularly suspicious of the ALDOC and MCT4 data in which genes that are purportedly induced <100-fold with sh1 against FILNC1 and then can be suppressed back to a level virtually indistinguishable from the starting point with only a very modest reduction in MYC (Fig. 4D). As with Fig. 3B, protein levels should be determined in addition to RNA.

We thank the reviewer for asking this question. Rebuttal letter **Figure 6A** presents the original figure this question

Figure 6. The effect of Myc knockdown in FILNC1 deficient cells on the expression of metabolic enzyme genes. *: P<0.05; **: P<0.01; ***: P<0.001.

refers to. Please note that, to incorporate the data from different metabolic enzyme genes with different fold changes into one bar graph panel, here we used log 2 scale (2, 4, 8, 16 etc) in y-axis, which accounts for the unnaturally narrow error bars for which the reviewer expressed concern. Rebuttal letter **Figure 6B** re-plotted all the data using separate bar graphs with linear y-axis scale, which shows regular error bars. In our revised manuscript, we presented data using individual bar graphs (Fig. 4E).

Fig. 5A: AUF1 has been described in many studies as a MYC mRNA binding factor, including the Liao/2007 paper that the authors reference. In particular, Liao/2007 identified MYC mRNA bound to AUF1 in a RIP experiment, and previous studies showed an interaction between AUF1 and MYC mRNA using in vitro binding experiments. Furthermore, the methods for the current study state that cell lysates were incubated with biotinylated RNA overnight, then RNA-protein complexes were isolated and subjected to mass spec. Were these experiments conducted under specific conditions of glucose starvation?

We thank the reviewer for asking this question. In the method of our revised manuscript (page 17, under “RNA pull-down assays and mass spectrometry”), we have clarified that these experiments were conducted using the lysates collected from cells that had been cultured under 1 mM glucose condition. In our revised manuscript, we have further clarified this in the corresponding text description (see the last paragraph in page 9 “we performed RNA-pulldown experiments followed by mass spectrometry to identify *FILNC1* interacting proteins under glucose starvation condition”).

Given the published data on AUF1 interaction with the MYC UTR and the protocol described in the manuscript, there is no explanation for why the authors do not see an interaction between the MYC UTR and AUF1 in vitro in the +glucose lane (Fig. 5A, lane 5). This is very contrary to published data. Furthermore, the supposed differential binding between +glucose and -glucose extracts is too striking to be believable (Fig. 5A: compare lanes 4 and 5 to lanes 7 and 8). Finally, why does AUF1 bind to FILNC1 with +glucose in Fig. 5B but not 5A? These data don't make sense.

Figure 7. RNA pulldown to examine binding between AUF1 and FILNC1 or MycUTR.

We thank the reviewer for asking this question. In our original Figure 5A, a short exposure (SE) of AUF1 western blotting was shown. In our revised manuscript (see rebuttal letter **Figure 7**, now Fig 5A in the revised manuscript), we also included a long exposure (LE) of the same AUF1 blotting, which revealed the binding between AUF1 protein and MycUTR (as well as FILNC1) under 25 mM glucose condition. Thus, we acknowledge that there is AUF1 protein-Myc mRNA interaction under normal culture condition. We hope the reviewer would agree that the glucose starvation-induced binding between AUF1 and Myc mRNA (or FILNC1) is intriguing, and provides important mechanistic insight on our studies.

We should point out that (1) the previous study on AUF1 protein-Myc mRNA interaction was conducted in different cell lines from the one used in this study (786-O cells), and it is likely that the exact binding strength between AUF1 protein and Myc mRNA might be varied among different cell lines, which is very common in protein-protein or protein-RNA interactions; and (2) Our study from both RNA pull down (rebuttal letter **Figure 7**) and RIP assays (rebuttal letter **Figure 8**) support the model that, at least in the cell line used in our study, there may be relatively weak interaction between FILNC1 and AUF1 under 25 mM glucose condition, and glucose starvation significantly increased FILNC1-AUF1 interaction. However, RNA pulldown and RIP are two different assays to study protein-RNA interaction with different sensitivities and limitations; thus, it is often difficult to directly compare the exact fold change between these two assays.

Fig. 5B: The authors state that the data is presented for AUF binding to FILNC1 after normalization of FILNC1 input level. I don't understand the logic in normalizing. If there's more FILNC1 bound to AUF per cell, then these data should be presented along with quantitation of RNA levels.

We thank the reviewer for this nice suggestion to further improve the clarity of our data. Here we show the data as suggested by this reviewer (rebuttal letter **Figure 8**, now Fig. 5B in the revised manuscript), which shows that (1) glucose starvation increased FILNC1 input level (consistent with the data shown in Figure 1 in our manuscript that FILNC1 is induced upon glucose starvation); and (2) glucose starvation resulted in a further fold increase of the FILNC1 level in AUF1 precipitates compared with that in FILNC1 input level, indicating that glucose starvation also increased the binding between FILNC1 and AUF1 protein.

Fig. 5D: To this reviewer's eye, the ratio of lanes 1 and 2 (siCon/siAUF1) is the same as lanes 3 and 4 (siCon/siAUF1 with additional sh1 against shRNA FILNC1). Why are no data shown for +glucose? If AUF has a constant impact on MYC protein level regardless of cell state, there may be no specificity to FILNC. See comments on Fig. 5F below.

We thank the reviewer for asking this excellent question. This question refers to the data presented in the left panel of rebuttal letter **Figure 9**. Here we addressed the question whether, under glucose starvation condition, the effect of FILNC1 knockdown (sh1) on Myc protein level is mediated through AUF1. The approach to address this question is to examine whether the effect of FILNC1 knockdown on Myc level is at least partially abolished in AUF1 knockdown (siAUF1) condition under glucose starvation. We want to clarify two important points here: (1) as already explained above, FILNC1 knockdown mainly affects Myc protein level under glucose starvation (1 mM glucose) condition, thus here we focus on analysis under glucose starvation; (2) here we need to compare **the ratio of lane 3/lane 1** (Myc protein level change upon FILNC1 knockdown under AUF1 proficient condition) and **the ratio of lane 4/lane 2** (Myc protein level change upon FILNC1 knockdown under AUF1 deficient condition).

Figure 8. RIP assay to examine AUF1 binding with FILNC1 under 25 and 1 mM glucose. ***: P<0.001.

Figure 9. The effect of FILNC1 knockdown on Myc protein level under glucose starvation is at least partially mediated through AUF1. *: P < 0.05.

To address this question, we have quantified c-Myc protein level normalized with Vinculin protein level from three independent experiments. As shown in the right panel of rebuttal letter **Figure 9**, The analysis revealed that the ratio of lane 3/lane 1 is **2.9**, while the ratio of lane 4/lane 2 is **1.5** (which means, under glucose starvation condition, FILNC1 knockdown increased Myc level by 2.9 fold under AUF1 proficient condition, while only increased Myc by 1.5 fold under AUF1 deficient condition). These data suggest that the effect of FILNC1 knockdown on Myc level is at least partially abolished in AUF1 knockdown condition. We should point out that, as shown in AUF1 Western blotting, siAUF1 only achieved partial knockdown of AUF1, thus it

is likely that our data is underestimated. In any event, even with this underestimated data, our results clearly show that the effect of FILNC1 knockdown on Myc protein level is at least partially mediated through AUF1. In our revised manuscript we have incorporated these new data in Fig. 5E, and emphasized this conclusion in the corresponding text description (the last paragraph, page 10).

Fig. 5F: The experiment in this figure offered the authors an opportunity to directly test their proposed model, i.e. that FILNC1 titrates AUF protein away from the MYC mRNA and hence suppresses MYC expression. This figure should have presented a straightforward assessment of the amount of MYC mRNA bound to AUF1 in a RIP experiment +/- glucose, with careful quantitation of all mRNA and protein levels, including FILNC1. These data should also be presented with careful normalization to levels per cell. Instead, the authors present a figure with a complex set of binding ratios that obscure the actual level of RNA-protein binding. This presentation is both confusing and illogical, and it does not directly address their model.

We thank the reviewer for asking this question to further improve the clarity of our data. Rebuttal letter **Figure 10** (which is presented as Fig. 5G in the revised manuscript) presents the data requested by this reviewer, which shows the FILNC1 expression level (panel A) and Myc level as input or upon AUF RIP (panel B) in control shRNA vs FLINC1 shRNA-infected cells under both 25 mM and 1 mM glucose conditions. The data shows that FILNC1 knockdown did not affect AUF1 protein binding with Myc mRNA under 25 mM glucose, but increased AUF1-Myc mRNA interaction under 1 glucose condition, which is in line with our proposed model.

Figure 10. FILNC1 expression level (A) and Myc level as input or upon AUF RIP (B) in control shRNA vs FLINC1 shRNA-infected cells under both 25 mM and 1 mM glucose conditions. *: P<0.05; **: P<0.01.

Fig. 6 presents complex correlative data and the results are barely within the range of statistical significance.

We appreciate the question from this reviewer. Figure 6 of our manuscript presents the correlative data from analyses of human cancer patients and provides additional support to our extensive mechanistic studies from in vitro and in vivo analyses. For all the analyses presented in this figure, we have conducted appropriate statistical analyses (as described in method), and the results are clearly statistically significant (such as P < 0.05).

Reviewers' comments:

Reviewer #3 (Remarks to the Author):

Xiao et al. present a revised manuscript describing experiments aimed at the hypothesis that a noncoding RNA, called FILNC1, is specifically induced under low glucose growth conditions in renal cancer. The authors have addressed some of the concerns I raised in my previous review but they still ignore fundamental issues that don't make sense. I'll focus on four issues although there are many others.

1) Another reviewer and I requested that the authors provide data on Myc protein stability under the conditions of their experiments. They provide figure 4 in which western blot data are presumably scanned and quantitated to generate a decay curve which is fitted to a LINEAR curve. The authors apparently do not understand that, when plotted on linear axes, a decay curve will form an exponential curve, NOT linear. From the western blots, Myc protein half-life is much less than the 32-37min that they claim. Given the mishandling of these data, it makes me skeptical of many of the other experiments.

2) In their revised Fig. 5, they show that there are equal levels of AUF1 protein under 25mM and 1mM glucose conditions, yet AUF1 does not bind to either FILNC1 or the MycUTR in an in vitro assay. However, AUF1 binds to both FILNC1 and the MycUTR under 1mM glucose. Why? Is AUF1 modified under 25mM glucose so it doesn't bind to RNA at all, or is it in a complex with some other RNA? This doesn't explain why a relatively small change in FILNC1 expression (~3-fold in manuscript Fig. 1C) only affects Myc protein level in 1mM glucose.

3) I raised a concern about the reported induction of some of the metabolic genes by 100-fold or greater. Their response is that some RNAs are expressed at such low levels so that any induction gives a huge fold-change. Yet the change in Myc protein level is 3-fold at best (Fig. 4D). The author's data are that this small change on Myc level induces 100-150 fold changes in ALDOC and MCT4 mRNAs (Fig. 4E). There have been hundreds of publications on mRNA changes in response to Myc levels, and NONE have ever documented such huge relative changes in mRNA, regardless of expression level.

4) The authors base their claims on the presumption that small changes in FILNC1 expression (3-fold) can induce huge changes in metabolic enzyme mRNA. They test this in Fig 6 but they used >10,000-fold over expression (Fig. 6A). How can the authors base any conclusions on such massive overexpression?

Detailed Point-by-point response to the reviewer's comments:

Reviewer #3:

Xiao et al. present a revised manuscript describing experiments aimed at the hypothesis that a noncoding RNA, called FILNC1, is specifically induced under low glucose growth conditions in renal cancer. The authors have addressed some of the concerns I raised in my previous review but they still ignore fundamental issues that don't make sense. I'll focus on four issues although there are many others.

We thank the reviewer for taking time and effort to review our manuscript and to help us strengthen our manuscript. I hope this reviewer will be satisfactory with our further revision with additional data and clarification.

1) Another reviewer and I requested that the authors provide data on Myc protein stability under the conditions of their experiments. They provide figure 4 in which western blot data are presumably scanned and quantitated to generate a decay curve which is fitted to a LINEAR curve. The authors apparently do not understand that, when plotted on linear axes, a decay curve will form an exponential curve, NOT linear. From the western blots, Myc protein half-life is much less than the 32-37min that they claim. Given the mishandling of these data, it makes me skeptical of many of the other experiments.

We thank the reviewer for pointing out this important issue. We agree with the reviewer that this is our oversight. In our original analysis, the western blotting data at different time points were fitted into a linear curve. As kindly instructed, we have reanalyzed our data from three independent experiments, and now presented the protein decay curve using log₂ of quantitated western blotting data, which typically exhibits linear curves, as shown in many other publications in the literature [for example, see “Wang, Z. *et al.* SCF^{β-TRCP} promotes cell growth by targeting PR-Set7/Set8 for degradation. *Nat. Commun.* 6:10185 (2015)” and “Arabi, A. *et al.* Proteomic screen reveals Fbw7 as a modulator of the NF-κB pathway. *Nat. Commun.* 3:976 (2012)”]. As shown in the **rebuttal letter Figure 1**, the new analysis confirmed that FILNC1 knockdown did not significantly affect c-Myc protein half-life

Figure 1. FILNC1 knockdown does not affect c-Myc protein stability under 1 mM glucose condition.

under 1 mM glucose (such as $P > 0.05$). Notably, c-Myc protein half-lives from our analysis (25 to 30 min) are also in line with literature reports, which show that c-Myc protein half-life usually is 20 to 30 min, validating our CHX analysis and half-life calculation methods [for example, see “Hann S. R., Eisenman R. N. Proteins encoded by the human c-myc oncogene: differential expression in neoplastic cells. *Mol. Cell. Biol.* 4:2486–2497 (1984)” and “Ramsay G., Evan G. I., Bishop J. M. The protein encoded by the human proto-oncogene c-myc. *Proc. Natl. Acad. Sci. USA* 81:7742–7746 (1984).”].

2) In their revised Fig. 5, they show that there are equal levels of AUF1 protein under 25mM and 1mM glucose conditions, yet AUF1 does not bind to either FILNC1 or the MycUTR in an in vitro assay. However, AUF1 binds to both FILNC1 and the MycUTR under 1mM glucose. Why? Is AUF1 modified under 25mM glucose so it doesn't bind to RNA at all, or is it in a complex with some other RNA?

We thank the reviewer for asking this insightful question. Our studies from both RNA pull down (Figure 5A) and RIP assays (Figure 5B) support the model that there is relatively weak interaction between FILNC1 and AUF1 under 25 mM glucose condition, and glucose starvation significantly increases FILNC1-AUF1 interaction. The exact mechanism by which glucose starvation regulates FILNC1 interaction with AUF1 remains less clear. As nicely suggested by this reviewer, it is possible that glucose starvation induces post-translational modification (such as phosphorylation) of AUF1, which may affect FILNC1-AUF1 interaction. Alternatively, glucose starvation may also affect the post-transcriptional modification (such as RNA methylation) on FILNC1 lncRNA, which may further influence FILNC1 binding to AUF1. It will be interesting to test these hypotheses in the future studies. We have incorporated this discussion in the second paragraph, page 14. We hope the reviewer would agree that, considering the scope and space limit for publications in *Nature Communications*, a complete understanding of the regulation of AUF1-FILNC1 interaction by glucose starvation will be beyond the scope of the current manuscript, which has already presented large amount of data (including 7 main figures, 15 supplementary figures, and 1 table).

This doesn't explain why a relatively small change in FILNC1 expression (~3-fold in manuscript Fig. 1C) only affects Myc protein level in 1mM glucose.

We believe that our data indeed provide mechanistic insight on why FILNC1 only affects Myc protein level under glucose starvation condition. Our data suggest that glucose starvation regulates FILNC1-AUF1-c-Myc signaling axis via at least two mechanisms: First, glucose starvation induces FILNC1 transcription, which is at least partly mediated by FoxO transcription factors. Second, glucose starvation also promotes FILNC1-AUF1 interaction (note that, in our in vitro binding assay, we used the same amount of in vitro synthesized FILNC1, and showed that there is still increased interaction between FILNC1 and AUF1). Thus, under glucose starvation,

through combinatorial effects of increases in both FLINC1 levels and FLINC1-AUF1 interaction, more AUF1 is sequestered by FILNC1 from binding c-Myc mRNA, leading to down-regulation of c-Myc protein levels. We have incorporated this discussion in the second paragraph, page 14.

3) I raised a concern about the reported induction of some of the metabolic genes by 100-fold or greater. Their response is that some RNAs are expressed at such low levels so that any induction gives a huge fold-change. Yet the change in Myc protein level is 3-fold at best (Fig. 4D). The author's data are that this small change on Myc level induces 100-150 fold changes in ALDOC and MCT4 mRNAs (Fig. 4E). There have been hundreds of publications on mRNA changes in response to Myc levels, and NONE have ever documented such huge relative changes in mRNA, regardless of expression level.

We thank the reviewer for asking the question. The dramatic fold changes of MCT4 and ALDOC upon FILNC1 or cMyc knockdown likely reflect the low basal expression of these genes in control shRNA cells, as judged by high Ct values from real time PCR analysis; under the backdrop of this low basal expression, we consistently observed that FILNC1 knockdown significantly increased the expression levels of these two genes (ALDOC Ct values: 34.3 in shLuc, 28.4 in sh1, and 28.7 in sh2; MCT4 Ct value: 35.1 in shLuc, 27.9 in sh1, and 28 in sh2). As shown in the **rebuttal letter Figure 2A and 2B** (Supplementary Figure 6 and Supplementary Figure 12 in our revised manuscript), we now provide the western blotting data of MCT4 and ALDOC to the corresponding conditions, which confirmed the low basal expression levels of these genes in control shRNA cells, dramatic increases of protein levels in FILNC1 knockdown cells under 1 mM glucose, and normalization to basal levels upon Myc knockdown in FILNC1 knockdown cells. We hope our new data provide additional important evidence to convince this reviewer. We are also open to the suggestions from this reviewer on how to present our real-time PCR data in a better way.

Figure 2. FILNC1 knockdown increased protein levels of ALDOC and MCT4 under 1 mM glucose (A) and knocking down c-Myc normalized ALDOC and MCT4 protein levels in FILNC1 knockdown cells under 1 mM glucose (B).

4) The authors base their claims on the presumption that small changes in FILNC1 expression (3-fold) can induce huge changes in metabolic enzyme mRNA. They test this in Fig 6 but they used >10,000-fold over expression (Fig. 6A). How can the authors base any conclusions on such massive overexpression?

We thank the reviewer for asking the question. The data refers to the figure presented in the **rebuttal letter Figure 3**. In this experiment, we chose a renal cancer cell line with FILNC1 low expression and then established a renal cancer cell line with stable overexpression of FILNC1. This is a commonly used strategy in gain-of-function studies. Here we measured FILNC1 level based on real time PCR. The huge increase of FILNC1 level upon overexpression partly reflects the extremely low expression level of endogenous FILNC1 in this cell line (Ct value 38.4 in control cells vs Ct value 22.6 in FILNC1 overexpressing cells). We agree with the concerns for gain-of-function studies with massive overexpression; We want to emphasize that the expression level of overexpressed FILNC1 in 769P cells (Ct value 22.6) is within reasonable range to the level of endogenous FILNC1 in 786O cells under glucose starvation (Ct value 24.3), under which we performed knockdown experiments, confirming that the FILNC1 expression level used in overexpression studies is of physiological relevance. Importantly, we also presented large amount of data based on shRNA-mediated knockdown experiments in our studies. These different approaches, each with its own limitations, together provide complimentary data to support our conclusion.

We should mention that, in our original manuscript, we presented this figure with FILNC1 overexpression as supporting data in the supplementary figure, but reviewer #1 commented that *“The overexpression data are important and should be inserted to the main figures but not just presented as an online figure”*. In the revised manuscript, we took this suggestion and presented it as Figure 6.

Figure 3. FILNC1 stable overexpression in 769P renal cancer cells.

Reviewers' comments:

Reviewer #6 (Remarks to the Author): Recruited to comment on the authors rebuttal to Reviewer #3

1) Another reviewer and I requested that the authors provide data on Myc protein stability under the conditions of their experiments. They provide figure 4 in which western blot data are presumably scanned and quantitated to generate a decay curve which is fitted to a LINEAR curve. The authors apparently do not understand that, when plotted on linear axes, a decay curve will form an exponential curve, NOT linear. From the western blots, Myc protein half-life is much less than the 32-37min that they claim. Given the mishandling of these data, it makes me skeptical of many of the other experiments.

Here the reviewer is wrong. The curve is linear, since the y-axis is an exponential axis, so the data handling is absolutely correct. Looking at the data, I would concur with the authors conclusions.

2) In their revised Fig. 5, they show that there are equal levels of AUF1 protein under 25mM and 1mM glucose conditions, yet AUF1 does not bind to either FILNC1 or the MycUTR in an in vitro assay. However, AUF1 binds to both FILNC1 and the MycUTR under 1mM glucose. Why? Is AUF1 modified under 25mM glucose so it doesn't bind to RNA at all, or is it in a complex with some other RNA? This doesn't explain why a relatively small change in FILNC1 expression (~3-fold in manuscript Fig. 1C) only affects Myc protein level in 1mM glucose.

This comment is correct. But one could argue - and I would subscribe to that - that the detailed mechanism of how the complex is regulated in vivo may be beyond the scope of this initial report.

3) I raised a concern about the reported induction of some of the metabolic genes by 100-fold or greater. Their response is that some RNAs are expressed at such low levels so that any induction gives a huge fold-change. Yet the change in Myc protein level is 3-fold at best (Fig. 4D). The author's data are that this small change on Myc level induces 100-150 fold changes in ALDOC and MCT4 mRNAs (Fig. 4E). There have been hundreds of publications on mRNA changes in response to Myc levels, and NONE have ever documented such huge relative changes in mRNA, regardless of expression level.

This comment is correct. I agree with the reviewer that Figure 4E reports unusually high expression changes for Aldoc, Pdk4 and Mct4. Comparison of these data with published data (e.g. from a doxycycline-repressible T-ALL model, which has generated probably the most MYC-dependent cells ever - from Felsher and colleagues) suggests that the values are too high. But to take this as an argument that the data are wrong, goes too far in my view. I would request that the authors express an siRNA-resistant MYC and use this to confirm the specificity of the siRNA effect. Also, the concern that these large values come from very low baseline expression (so there is essentially a division by zero) should be alleviated by an immunoblot showing that the proteins are expressed at a detectable level.

4) The authors base their claims on the presumption that small changes in FILNC1 expression (3- >10,000-fold over expression (Fig. 6A). How can the authors base any conclusions on such used massive overexpression?

This comment is also correct, since the effects on MYC in this setting are quite weak. I think the authors need to concede that regulation of FLINC levels is only part of the mechanism (see also comment 2)- this may already be in text. A very good experiment would be to show that ectopic expression a MYC mRNA without the AUF binding, but not one that is sensitive to FLINC1 rescues expression of target genes.

Detailed Point-by-point response to the reviewer's comments:

Reviewer #6 (Remarks to the Author):

We thank the reviewer for taking time and effort to review our manuscript and to help us strengthen our manuscript. We hope this reviewer will be satisfactory with our manuscript revision with additional data and clarification.

1) Another reviewer and I requested that the authors provide data on Myc protein stability under the conditions of their experiments. They provide figure 4 in which western blot data are presumably scanned and quantitated to generate a decay curve which is fitted to a LINEAR curve. The authors apparently do not understand that, when plotted on linear axes, a decay curve will form an exponential curve, NOT linear. From the western blots, Myc protein half-life is much less than the 32-37min that they claim. Given the mishandling of these data, it makes me skeptical of many of the other experiments.

Reviewer #6 comment: Here the reviewer is wrong. The curve is linear, since the y-axis is an exponential axis, so the data handling is absolutely correct. Looking at the data, I would concur with the author's conclusions.

We thanks the reviewer for the positive comment.

2) In their revised Fig. 5, they show that there are equal levels of AUF1 protein under 25mM and 1mM glucose conditions, yet AUF1 does not bind to either FILNC1 or the MycUTR in an in vitro assay. However, AUF1 binds to both FILNC1 and the MycUTR under 1mM glucose. Why? Is AUF1 modified under 25mM glucose so it doesn't bind to RNA at all, or is it in a complex with some other RNA? This doesn't explain why a relatively small change in FILNC1 expression (~3-fold in manuscript Fig. 1C) only affects Myc protein level in 1mM glucose.

Reviewer #6 comment: This comment is correct. But one could argue - and I would subscribe to that - that the detailed mechanism of how the complex is regulated in vivo may be beyond the scope of this initial report.

We thank the reviewer for agreeing that the detailed mechanism of the regulation of AUF1-FILNC1 interaction by glucose starvation will be beyond the scope of the current manuscript.

3) I raised a concern about the reported induction of some of the metabolic genes by 100-fold or greater. Their response is that some RNAs are expressed at such low levels so that any induction gives a huge fold-change. Yet the change in Myc protein level is 3-fold at best (Fig. 4D). The author's data are that this small change on Myc level induces 100-150 fold changes in ALDOC and MCT4 mRNAs (Fig. 4E). There have been hundreds of publications on mRNA changes in response to Myc levels, and NONE have ever documented such huge relative changes in mRNA, regardless of expression level.

Reviewer #6 comment: This comment is correct. I agree with the reviewer that Figure 4E reports

unusually high expression changes for Aldoc, Pdk4 and Mct4. Comparison of these data with published data (e.g. from a doxycycline-repressible T-ALL model, which has generated probably the most MYC-dependent cells ever - from Felsher and colleagues) suggests that the values are too high. But to take this as an argument that the data are wrong, goes too far in my view. I would request that the authors express an siRNA-resistant MYC and use this to confirm the specificity of the siRNA effect.

We thank the reviewer for the comment and suggestions to further improve our manuscript. We want to emphasize and hope the reviewer would agree that Myc regulation of transcriptional targets, including fold changes of Myc targets, is very context- and cell line-dependent. We have also performed the experiments requested by this reviewer. As shown in the **rebuttal letter Figure 1** (Supplementary Figure 13 in our revised manuscript), restoring siRNA-resistant Myc in FILNC1/Myc double knockdown cells largely normalized the downregulation of expression levels of metabolic enzyme genes, including ALDOC and MCT4, caused by Myc knockdown, thus confirming the specificity of Myc siRNA effect.

Figure 1. Restoring Myc in Myc knockdown cells largely normalized the expression levels of glucose metabolism genes caused by Myc knockdown. sh1: FILNC1 shRNA #1; siMyc: Myc siRNA; siCon: Control siRNA; Empty: empty vector. *: P<0.05; **: P<0.01; ***: P<0.001.

Also, the concern that these large values come from very low baseline expression (so there is essentially a division by zero) should be alleviated by an immunoblot showing that the proteins are expressed at a detectable level.

We thank the reviewer for this great suggestion. The **rebuttal letter Figure 2** (Supplementary Figure 6 in our revised manuscript) shows both short and long exposure of ALDOC and MCT4 western blotting in control and FILNC1 knockdown cells.

Figure 2. FILNC1 knockdown increased protein levels of ALDOC and MCT4 under 1 mM glucose. SE: short exposure; LE: long exposure.

4) The authors base their claims on the presumption that small changes in *FILNC1* expression (3-fold) can induce huge changes in metabolic enzyme mRNA. They test this in Fig 6 but they used >10,000-fold over expression (Fig. 6A). How can the authors base any conclusions on such massive overexpression?

Reviewer #6 comment: This comment is also correct, since the effects on MYC in this setting are quite weak. I think the authors need to concede that regulation of FLINC levels is only part of the mechanism (see also comment 2)- this may already be in text. A very good experiment would be to show that ectopic expression a MYC mRNA without the AUF binding, but not one that is sensitive to FLINC1 rescues expression of target genes.

We thank the reviewer for the comment and great suggestions to further improve our manuscript. We agree with this reviewer that *FILNC1* regulation of energy metabolism and renal tumor development likely involves different mechanisms, but our studies show that Myc is at least one downstream effector to mediate the biological effects afforded by *FILNC1* deficiency under energy stress (see the last sentence in the second paragraph, page 9 in our manuscript). We also acknowledge that Myc expression is controlled by multiple mechanisms (see the second paragraph, page 3 in our manuscript), and *FILNC1* regulation of Myc is part of the mechanisms.

We also conducted the experiments suggested by this reviewer. The **rebuttal letter Figure 3** (Supplementary Figure 17 in our revised manuscript) shows that re-expression of a Myc (using a Myc construct without Myc 3'UTR region, thus escaping *FILNC1* regulation of Myc through AUF1) in *FILNC1*-overexpressing cells to the level comparable to that in control cells rescued the expression of target genes.

Finally, as shown in the **rebuttal letter Figure 3** (Supplementary Figure 16 in our revised

Figure 3. Re-expression of a Myc in *FILNC1*-overexpressing cells restored the expression of target genes. *: P<0.05; **: P<0.01.

	786O			769P+ FILNC1		
GAPDH	15.04	15.8	15.77	15.68	15.51	15.7
FILNC1	24.21	24.71	24.04	22.85	21.95	23.09

Figure 3. Comparison of expression levels of overexpressed *FILNC1* in 769P cells and endogenous *FILNC1* in 786O cells.

manuscript), the expression level of overexpressed FILNC1 in 769P cells was within reasonable range to that of endogenous FILNC1 in 786-O cells (a cell line with high expression of FILNC1) under glucose starvation, suggesting that the FILNC1 expression level used in overexpression studies is of physiological relevance. Together, we hope our clarification and new data can alleviate the concern from this reviewer on our overexpression studies.

REVIEWERS' COMMENTS:

Reviewer #1 (Remarks to the Author):

I agree both comment point 3 and point 4 from the previous reviewers (reviewer #3 and reviewer #6). The changes of some genes showed in Figure 4 are unexpected and unusual. In most cases, the fold changes of MYC targeted genes are much lower, since MYC may serve as a universal transcription amplifier (but not a gene specific transcriptional factor) in cancer cells (Cell. 2012 Sep 28;151(1):68-79. doi: 10.1016/j.cell.2012.08.033.; and Cell. 2012 Sep 28;151(1):56-67. doi: 10.1016/j.cell.2012.08.026.). Therefore, based on the data showed in Figure 4, MYC-independent mechanism(s) might be also involved in the regulatory loop(s). I would like to suggest the authors carefully discuss this issue in the discussion section. Additionally, they need to add their new data showed in the response letter to the final manuscript.

Detailed Point-by-point response to the reviewer's comments:

Reviewer #1 (Remarks to the Author):

I agree both comment point 3 and point 4 from the previous reviewers (reviewer #3 and reviewer #6). The changes of some genes showed in Figure 4 are unexpected and unusual. In most cases, the fold changes of MYC targeted genes are much lower, since MYC may serve as a universal transcription amplifier (but not a gene specific transcriptional factor) in cancer cells (Cell. 2012 Sep 28;151(1):68-79. doi: 10.1016/j.cell.2012.08.033.; and Cell. 2012 Sep 28;151(1):56-67. doi: 10.1016/j.cell.2012.08.026.). Therefore, based on the data showed in Figure 4, MYC-independent mechanism(s) might be also involved in the regulatory loop(s). I would like to suggest the authors carefully discuss this issue in the discussion section. Additional, they need to add their new data showed in the response letter to the final manuscript.

We thank this reviewer for the insightful suggestions. We agree with this reviewer that the expression changes of some glycolysis genes (such as MCT4 and ALDOC) are much higher than fold changes of most Myc target genes. Thus, we cannot rule out the possibility that FILNC1 regulation of the expression of some glycolysis genes might involve Myc-independent mechanisms. We have added this discussion in the page 9, the 2nd paragraph of our revised manuscript. In addition, we have added all the new data shown in previous rebuttal letter to the final manuscript.